# TOWARDS AN AI MUSICIAN: SYNTHESIZING SHEET MUSIC PROBLEMS FOR MUSICAL REASONING

## ABSTRACT

Enhancing the ability of Large Language Models (LLMs) and Multimodal Large Language Models (MLLMs) to interpret sheet music is a crucial step toward building AI musicians. However, current research lacks both evaluation benchmarks and training data for sheet music reasoning. Drawing from mathematics, where simple operations can generate a boundless universe of verifiable problems, we introduce a novel approach that treats core music theory rules, such as those governing beats and intervals, as programmatic functions to systematically synthesize a vast and diverse corpus of sheet music reasoning problems. This approach allows us to introduce a data synthesis framework that generates verifiable sheet music questions in both textual and visual modalities, leading to the Synthetic Sheet Music Reasoning Benchmark (SSMR-Bench) and a complementary training set. Evaluation results on SSMR-Bench highlight the key role reasoning plays in interpreting sheet music, while also pointing out the ongoing challenges in understanding sheet music in a visual format. By leveraging synthetic data for RLVR, models like Qwen3-8B-Base and Qwen2.5-VL-7B-Instruct show significant improvements on the SSMR-Bench. Additionally, they also demonstrate considerable advancements on previously established human-crafted benchmarks, such as MusicTheoryBench and the music subset of MMMU. Finally, our results show that the enhanced reasoning ability can also facilitate music composition.

## 1 INTRODUCTION

*"Sheet music is the written language of musicians."*

Recent advancements in Large Language Models (LLMs) and Multimodal Large Language Models (MLLMs) have inspired researchers to explore the potential of developing AI musicians (Qu et al., 2025; Bradshaw & Colton, 2025; Wang et al., 2024). Given that sheet music is the universal language of musicians, the ability to read and interpret it is an essential step for AI musicians (Yuan et al., 2024; Wang et al., 2025). We term this capability sheet music reasoning. As illustrated in Figure 1, sheet music reasoning differs fundamentally from Music Knowledge QA (Li et al., 2024), which evaluates memorized knowledge, and from sheet music recognition (Chen et al., 2025a), which focuses on identifying notation from images. Instead, sheet music reasoning requires applying

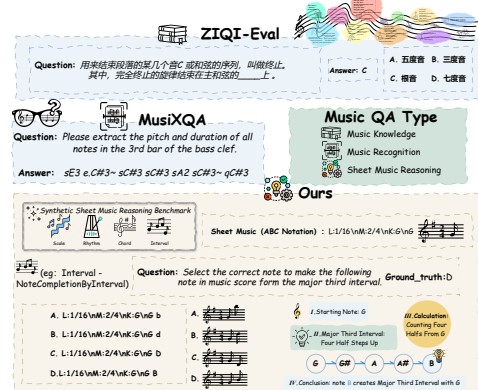

Figure 1: Comparison of Sheet Music Reasoning QA, Knowledge QA, and Music Recognition.

learned musical knowledge to actively interpret and analyze sheet music. This task demands not only accurate recognition of musical symbols but also a nuanced understanding of how these symbols interact within the broader context of musical knowledge.

However, research on sheet music reasoning remains limited, with few standardized evaluation benchmarks or training datasets available. For instance, ChatMusician (Yuan et al., 2024) intro-

duces MusicTheoryBench, a benchmark comprising 367 human-designed questions, some of which specifically focus on sheet music reasoning in ABC notation, a textual representation of sheet music. Similarly, MMMU (Yue et al., 2024) provides a benchmark for evaluating multimodal models across large-scale, multi-disciplinary tasks that require college-level knowledge and deliberate reasoning, including a subset of questions that involve sheet music reasoning in image format. However, these datasets are entirely human-crafted, which limits their scalability for training.

Drawing from mathematics, where simple operations can generate a boundless universe of verifiable problems, we propose to leverage the rules of music theory to programmatically generate a scalable dataset of verifiable reasoning problems based on sheet music. Such a resource is invaluable for creating robust evaluation benchmarks and for training models with verifiable rewards (DeepSeek-AI et al., 2025) to improve their reasoning abilities. The practical implementation of this concept is our novel data synthesis framework. This system is designed to generate questions with staff notation presented in both textual and visual formats and includes nine Question Template Classes, categorized into Rhythm, Chord, Interval, and Scale. We use this framework to build the Synthetic Sheet Music Reasoning Benchmark (SSMR-Bench). The benchmark includes an evaluation set with 1,600 textual and 1,600 visual QA pairs, and a larger training set with 8,000 pairs for each modality.

We evaluate SSMR-Bench using a range of state-of-the-art LLMs and MLLMs. For the textual QA task, which represents sheet music in ABC notation, Qwen3-32B demonstrates performance on par with Deepseek-V3, while Deepseek-R1 exceeds Deepseek-V3 by 18.69% overall. This highlights the model's enhanced reasoning capability in understanding sheet music. For the visual QA task, Gemini2.5-Pro achieves the best performance with an average score of 56.62%, surpassing Qwen2.5-VL-72B-Instruct, InternVL3-78B, and other models. However, this performance is considerably lower than Gemini2.5-Pro's results on the same questions presented in textual format, highlighting the challenges of interpreting sheet music from images.

We train a range of models on our synthetic dataset using Group Relative Policy Optimization (GRPO, Shao et al., 2024). Across different model families (Qwen, Yang et al., 2025 and Llama, Grattafiori et al., 2024), sizes (3B–8B), and modalities (text and image), all models achieve substantial improvements on SSMR-Bench. Notably, Qwen3-8B-Base improves by 13.06%, reaching performance comparable to GPT-4.1-mini, while Qwen2.5-VL-7B-Instruct exhibits stronger multimodal reasoning, surpassing Gemini2.5-Pro. Beyond the in-distribution benchmark, evaluation on external benchmarks such as MusicTheoryBench (Yuan et al., 2024) and MMMU Music (Yue et al., 2024), also demonstrates consistent improvement across all models. Interestingly, training on synthetic musical data also improves mathematical reasoning, underscoring the generality of the elicited capabilities. Moreover, our evaluations show that enhanced reasoning ability leads to improvement in sheet music composition, with the trained Qwen3-8B-Base outperforming Qwen3-8B-Thinking and demonstrating greater musical coherence and accuracy.

In conclusion, we summarize our key contributions as follows:

- We are the first to leverage music theory rules to programmatically synthesize verifiable sheet music problems, using them as both an evaluation benchmark and a training set for Reinforcement Learning with Verifiable Rewards (RLVR).
- We develop a data synthesis framework based on this idea, which is capable of generating verifiable sheet music questions with staff notation in both textual and visual modalities, and we provide SSMR-Bench along with a corresponding training set in both modalities.
- We conduct comprehensive experiments to demonstrate the potential of automatically synthesizing verifiable sheet music problems based on music theory rules. Training on this synthetic data enhances models' reasoning abilities in sheet music, and this improved ability also shows promise in facilitating music composition.

## 2 SYNTHESIZE MUSIC REASONING QUESTIONS

### 2.1 FUNDAMENTALS OF SHEET MUSIC

Music is traditionally written on a staff, a set of five lines where visual symbols represent the core elements of music theory. The vertical position of a notehead indicates its pitch, while its shape (e.g., hollow or filled, with a stem or flag) defines its rhythm, or duration. These visual cues are

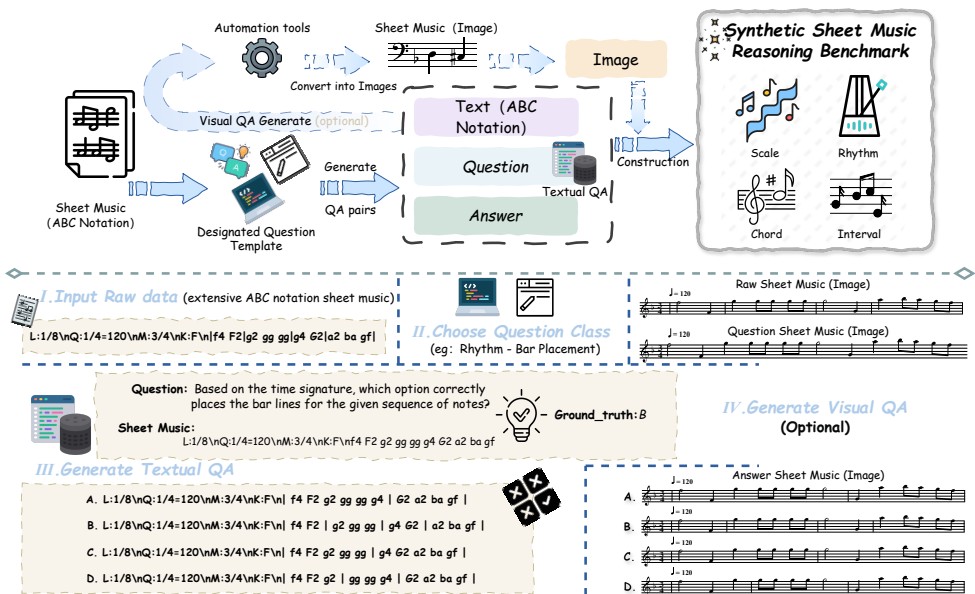

Figure 2: Overview of synthesizing verifiable sheet music questions. The upper plot illustrates the pipeline. Experts define rule-based Question Template Classes that generate questions and answers from existing sheet music in ABC notation, resulting in textual QA problems. The sheet music is then converted into an image, allowing the creation of Visual Question Answering (VQA)-style questions. The process is fully programmatic, without LLMs. The lower plot presents examples of generated textual and visual QA tasks. More details can be found in Appendix A.1.

organized into measures, separated by barlines, according to a time signature that dictates the underlying rhythmic pulse. While staff notation is intuitive for human musicians, its graphical nature is cumbersome for programmatic analysis. To bridge this gap, we utilize ABC notation, a text-based format that encodes the same musical information in a machine-readable way (Qu et al., 2025). In ABC, pitches are represented by letters (C, D, E), rhythmic values are specified with numerical modifiers, and measures are delineated by vertical bars (|). This format allows us to translate the abstract rules of music theory into parsable strings. Examples of ABC notation are provided in Appendix A.

## 2.2 DATA SYNTHESIS FRAMEWORK

Our framework is designed to generate questions that test a model's understanding of the fundamental music principles, from rhythmic calculation and chord identification to the analysis of intervals and scales. We outline the process of synthesizing sheet music questions based on music theory rules in Figure 2. Sheet music is first obtained in ABC notation. Then, the experts design specialized Question Template Classes derived from music theory rules, each capable of generating both questions and corresponding answers from the ABC representation. Applying these templates produces textual QA pairs. Subsequently, the sheet music in each question can be converted into images using *abcm2ps* and *ImageMagick*, enabling the construction of VQA questions. The entire pipeline is fully programmatic, without reliance on LLMs, ensuring controllability and interpretability.

**Example: Bar Placement Question Synthesis**

**Require:** $\mathcal{S}_{\text{orig}}$ (ABC notation)
1: **function** GENERATEQA($\mathcal{S}_{\text{orig}}$)
2:     $\mathcal{A}_{\text{corr}} \leftarrow \mathcal{S}_{\text{orig}}$
3:     $\mathcal{C}_{\text{txt}} \leftarrow$ REMOVEBARS($\mathcal{S}_{\text{orig}}$)
4:     $\mathcal{A}_{\text{incorr}} \leftarrow$ GENDISTRACTORS($\mathcal{C}_{\text{txt}}$)
5:     Choices $\leftarrow$ SHUFFLE($\{\mathcal{A}_{\text{corr}}\} \cup \mathcal{A}_{\text{incorr}}$)
6:     $\mathcal{Q}_{\text{txt}} \leftarrow (\mathcal{C}_{\text{txt}}, \text{Choices})$
7:     $\mathcal{Q}_{\text{vis}} \leftarrow$ RENDERIMGS($\mathcal{Q}_{\text{txt}}$)
8:     **return** ($\mathcal{Q}_{\text{txt}}, \mathcal{Q}_{\text{vis}}$)
9: **end function**

Figure 3: Algorithm for bar placement question synthesis, including bar removal, distractor generation, option shuffling, and rendering of textual and visual questions.

The lower plot in Figure 2 presents a visual illustration of an example generated by the Bar Placement Class. In this case, the bar lines of the original sheet music are removed, and the resulting incomplete score serves as input. According to music theory, the missing bar lines can be reconstructed by reasoning from the time signature, requiring models to infer rhythmic grouping and meter to restore the bar-line structure. Once the textual QA question is obtained, the sheet music in textual modality can be converted into images to generate the corresponding VQA question. Figure 3 provides the pseudocode for the Bar Placement Class, demonstrating how it functions.

## 2.3 DATA CONSTRUCTION

Table 1: Comparison of music-related QA datasets across different aspects, where checkmarks (✓) indicate presence and crosses (✗) indicate absence.

| Dataset | Sheet Music QA | Synthetic | Modality | | Trainable |
|---|---|---|---|---|---|
| | | | Textual | Visual | |
| MMMU (Yue et al., 2024) | ✓ | ✗ | ✗ | ✓ | ✗ |
| MusiXQA (Chen et al., 2025a) | ✗ | ✓ | ✗ | ✓ | ✓ |
| ZIQI-Eval (Li et al., 2024) | ✗ | ✗ | ✓ | ✗ | ✗ |
| MusicTheoryBench (Yuan et al., 2024) | ✓ | ✗ | ✓ | ✗ | ✗ |
| SSMR-Bench (Ours) | ✓ | ✓ | ✓ | ✓ | ✓ |

The scalable synthesis framework we designed currently provides 9 types of questions. All questions are categorized into four categories: Rhythm, Chord, Interval, and Scale, which are the fundamental elements of sheet music. Detailed descriptions of the question types and categories can be found in Appendix A. To solve these questions, LLMs and MLLMs need basic music knowledge, sheet music reading skills, and reasoning ability.

Through our data synthesis framework, we introduce the Synthetic Sheet Music Reasoning Benchmark (SSMR-Bench), comprising 1,600 textual and 1,600 visual questions. The sheet music used to generate questions is sampled from MelodyHub (Wu et al., 2024), a large-scale dataset in ABC notation. The distribution of question types and categories within the benchmark is illustrated in Figure 4. Furthermore, we offer a comprehensive training set, consisting of 8,000 textual and 8,000 visual questions. A comparative analysis between our framework and prior related benchmarks is presented in Table 1. Our framework produces both visual and textual modality questions for sheet music problems and constructs a verifiable training set without human curation, thereby enabling scalable data generation.

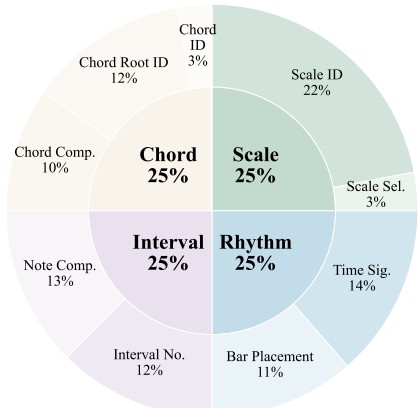

Figure 4: The composition of the SSMR-Bench test set consists of four categories: Rhythm, Chord, Interval, and Scale. Each category contributes equally, accounting for 25% of the total.

## 2.4 VERIFIABLE MUSIC TRAINING

Following the RLVR paradigm, we employ the Group Relative Policy Optimization (GRPO, Shao et al., 2024) algorithm for RL training. GRPO approximates the advantage of a trajectory by normalizing its reward against the mean and standard deviation of rewards from $N$ sampled solutions for a given query:

$$A_i = \frac{R(\tau_i) - \text{mean}(\{R(\tau_j) \mid j = 1, \ldots, N\})}{\text{std}(\{R(\tau_j) \mid j = 1, \ldots, N\})}, \tag{1}$$

where $R(\tau_i)$ is the reward for trajectory $\tau_i$. The rewards are calculated over $N$ trajectories sampled from the previous policy $\pi_{\theta_{\text{old}}}$. This relative scoring helps the model learn which outputs perform

better or worse than the group's average. Then, the GRPO objective is given by:

$$\mathcal{J}_{\text{GRPO}}(\theta) = \frac{1}{\sum_{i=1}^{N}|\tau_i|}\sum_{i=1}^{N}\sum_{t=1}^{|\tau_i|}\text{CLIP}(r_{i,t}(\theta), A_i, \epsilon) - \beta \cdot \mathbb{D}_{\text{KL}}[\pi_\theta \| \pi_{\text{ref}}]. \quad (2)$$

Here, $r_{i,t}(\theta) = \pi_\theta(\tau_{i,t}|q, \tau_{i,<t})/\pi_{\theta_{\text{old}}}(\tau_{i,t}|q, \tau_{i,<t})$ is the importance sampling ratio, adjusting for the fact that the trajectories were generated by the old policy $\pi_{\theta_{\text{old}}}$, ensuring unbiased policy updates.

## 3 EXPERIMENT

### 3.1 EXPERIMENTAL SETUP

**Evaluation.** We select the frontier LLMs and MLLMs to evaluate their performance on our SSMR-bench. For testing, the temperature is set to 0.7, with a length limit of 8192. We shuffle the multiple-choice options to avoid contamination. We utilize Math-Verify to parse the answers from the model responses and vLLM as the backend for inference serving.

**Implementation Details.** We employ multiple LLMs and MLLMs for reinforcement learning on our synthetic data. For the textual modality, we choose Qwen3-4B-Base, Qwen3-8B-Base, and Llama3.1-8B-IT (with "IT" referring to "Instruct"). For the visual modality, we select Qwen2.5-VL-7B-Instruct. Additionally, we include a variant of Qwen3-8B-Base trained on 8,000 math problems sampled from OpenR1 (Hugging Face, 2025) to compare the effectiveness of synthetic music reasoning data. Training is conducted using the GRPO algorithm with a batch size of 128, a rollout number of 8, and an update batch size of 64 over 300 training steps. The KL loss and entropy loss are removed. The reward function is based solely on accuracy, without incorporating format-related signals. The training template is provided in Appendix B.

### 3.2 RESULTS ON SYNTHETIC MUSIC REASONING BENCHMARK

Table 2 presents the performance of advanced models on SSMR-bench, covering both LLMs and MLLMs in the textual and visual modalities. The performance metrics reveal notable variations between different models, highlighting their strengths and limitations in each modality.

**Textual Modality Performance Comparison.** The results for textual QA clearly show that LLMs, such as GPT-5, outperform other models across all categories. Specifically, GPT-5 attains the highest overall accuracy at 97.00%, followed by Gemini-2.5-Pro at 96.10%, DeepSeek-R1 at 93.63%, and GPT-4.1 at 79.81%. In contrast, models like Qwen2.5-32B-IT and Llama3.1-70B-IT exhibit lower overall performance, with scores of 51.25% and 43.94%, respectively. Llama3.1-8B-IT shows the lowest performance, with an overall score of 20.94%.

**Performance of MLLMs on Visual Modality.** The performance of MLLMs on sheet music reasoning, with staff notation presented in image format, is substantially lower than that of LLMs on textual reasoning. The best-performing MLLM is Gemini 2.5 Pro, which is recognized as one of the most advanced models available. It achieves an overall accuracy of 55.44%, with particularly strong performance on Scale (84.75%). Other models, such as Qwen2.5-VL-72B-Instruct, InternVL3-78B, and InternVL2.5-78B, show accuracies of 45.56%, 45.75%, and 40.44%, respectively.

**Cross-Modality Performance Analysis.** The performance degradation observed in the visual modality stems from the prerequisite of recognition before reasoning. Unlike textual inputs that provide explicit semantic tokens, sheet music images require models to first accurately decode intricate visual patterns before any logical deduction can occur. Figure 5 illustrates the impact of this visual encoding stage by comparing Gemini-2.5-Pro and InternVL-2.5-78B across modalities. The results highlight distinct bottlenecks. Gemini-2.5-Pro suffers a sharp decline from 96.19% (textual) to 55.44% (visual), highlighting that its superior reasoning is bottlenecked by poor visual recognition. In contrast, InternVL-2.5-78B shows a narrow gap (46.06% vs. 40.44%), yet performs poorly in both settings. This suggests a core reasoning deficit rather than a visual issue, as the model fails to solve the underlying problems irrespective of input modality.

Table 2: Evaluation results of frontier models for textual and visual QA on the SSMR-bench. The table presents the performance of various models across different sheet music reasoning tasks, including Rhythm, Chord, Interval, and Scale, with the overall performance summarized for both textual and visual QA. "IT" is an abbreviation for "Instruct". The "Thinking" column indicates whether the reasoning model has the thinking mode activated.

| Models | Thinking | Synthetic Sheet Music Reasoning Benchmark | | | | |
| | | Rhythm (400) | Chord (400) | Interval (400) | Scale (400) | Overall (1600) |
|---|---|---|---|---|---|---|
| *Textual QA* | | | | | | |
| GPT-5 | ✓ | 97.75 | 97.50 | 92.75 | 100.00 | 97.00 |
| Gemini-2.5-Pro | ✓ | 98.25 | 96.50 | 90.25 | 99.75 | 96.19 |
| Deepseek-R1 | ✓ | 93.00 | 90.50 | 91.25 | 99.75 | 93.63 |
| GPT-4.1 | ✗ | 88.75 | 68.75 | 63.75 | 98.00 | 79.81 |
| Qwen3-32B-Thinking | ✓ | 80.00 | 75.00 | 62.25 | 89.50 | 76.69 |
| Deepseek-V3-250324 | ✗ | 78.75 | 71.75 | 58.75 | 90.50 | 74.94 |
| QwQ-32B | ✓ | 69.75 | 72.50 | 59.50 | 87.75 | 72.38 |
| GPT-4.1-mini | ✗ | 77.00 | 69.75 | 56.25 | 83.75 | 71.69 |
| Qwen3-8B-Thinking | ✓ | 54.00 | 60.50 | 50.75 | 66.25 | 57.88 |
| Qwen2.5-32B-IT | ✗ | 54.50 | 47.00 | 53.75 | 49.75 | 51.25 |
| GPT-4o-mini | ✗ | 48.00 | 54.50 | 48.50 | 40.25 | 47.81 |
| Llama3.1-70B-IT | ✗ | 39.75 | 41.00 | 47.00 | 48.00 | 43.94 |
| Llama3.1-8B-IT | ✗ | 23.50 | 19.00 | 18.00 | 23.25 | 20.94 |
| *Visual QA* | | | | | | |
| Gemini-2.5-Pro | ✓ | 44.75 | 61.25 | 32.25 | 83.50 | 55.44 |
| InternVL3-78B | ✗ | 34.25 | 44.75 | 29.00 | 75.00 | 45.75 |
| Qwen2.5-VL-72B-IT | ✗ | 39.75 | 41.75 | 27.50 | 73.25 | 45.56 |
| InternVL2.5-78B | ✗ | 29.00 | 31.00 | 32.25 | 69.50 | 40.44 |

**Effect of Reasoning on Performance.** The incorporation of reasoning capabilities significantly enhances model performance on music reasoning tasks. Notably, models like Qwen3-32B and QwQ-32B, with only 32 billion parameters, achieve results comparable to Deepseek-V3, which has 671 billion parameters. Moreover, Deepseek-R1, a reasoning model derived from Deepseek-V3, outperforms Deepseek-V3 by 18.69% in overall performance. This underscores the critical role of reasoning in boosting model accuracy and highlights that even models with fewer parameters can achieve competitive performance when equipped with advanced reasoning capabilities.

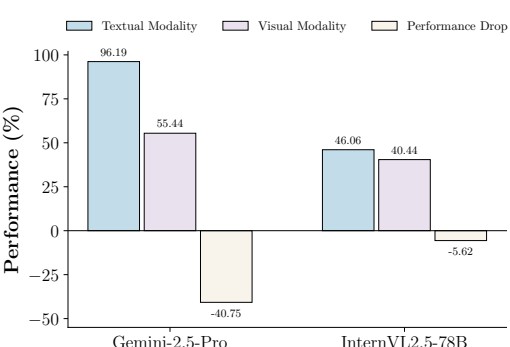

Figure 5: Performance degradation introduced by the visual modality.

### 3.3 RLVR ON SYNTHETIC MUSIC REASONING DATA

Compared to previous approaches, our data synthesis framework offers a simple yet effective method for scaling sheet music reasoning data with verifiable answers, enabling RLVR on both LLMs and MLLMs. We select various open-source LLMs and VLMs and train them on our synthetic dataset consisting of 8,000 examples. The LLMs, such as Qwen3-8B-Base, are trained on textual data, while the VLMs, like Qwen2.5-VL-7B-Instruct, are trained on visual data. All models are optimized using the GRPO algorithm. The results are presented in Table 3.

As shown in Table 3, all models show significant improvements on the SSMR-Bench after RL. For instance, in the Textual Modality, the Qwen3-8B-Base model initially achieved relatively low scores across all metrics, with an average of 23.18%. However, after applying GRPO to the synthetic

Table 3: Performance of Qwen3-4B-Base, Qwen3-8B-Base, and Llama3.1-8B-IT on Textual QA, and Qwen2.5-VL-3B-IT and Qwen2.5-VL-7B-IT on Visual QA tasks, after applying GRPO to synthetic data on the SSMR-bench. The table compares these results with their baseline models (without GRPO) and the corresponding Thinking models (if applicable).

| Models | Synthetic Sheet Music Reasoning Benchmark | | | | |
| --- | --- | --- | --- | --- | --- |
| | Rhythm | Chord | Interval | Scale | Overall |
| *Textual QA* | | | | | |
| Qwen3-4B-Base | 14.50 | 13.00 | 8.25 | 10.25 | 11.50 |
| +GRPO (Ours) | $85.50^{(+71.00)}$ | $50.50^{(+37.50)}$ | $58.25^{(+50.00)}$ | $81.50^{(+71.25)}$ | $68.94^{(+57.44)}$ |
| Qwen3-4B-Thinking | 54.00 | 53.50 | 48.75 | 55.50 | 52.94 |
| Qwen3-8B-Base | 23.25 | 32.25 | 18.25 | 19.00 | 23.18 |
| +GRPO (Ours) | $73.75^{(+50.50)}$ | $59.00^{(+26.75)}$ | $62.00^{(+43.75)}$ | $89.00^{(+70.00)}$ | $70.94^{(+47.76)}$ |
| Qwen3-8B-Thinking | 54.00 | 60.50 | 50.75 | 66.25 | 57.88 |
| Llama3.1-8B-IT | 23.50 | 19.00 | 18.00 | 23.25 | 20.94 |
| +GRPO (Ours) | $83.25^{(+59.75)}$ | $54.50^{(+35.50)}$ | $55.25^{(+37.25)}$ | $90.00^{(+66.75)}$ | $70.75^{(+49.81)}$ |
| *Visual QA* | | | | | |
| Qwen2.5-VL-3B-IT | 45.00 | 40.50 | 28.50 | 56.00 | 42.50 |
| +GRPO (Ours) | $51.00^{(+6.00)}$ | $54.75^{(+14.25)}$ | $40.50^{(+12.00)}$ | $89.75^{(+33.75)}$ | $59.00^{(+16.50)}$ |
| Qwen2.5-VL-7B-IT | 44.00 | 40.75 | 35.25 | 47.25 | 41.81 |
| +GRPO (Ours) | $66.50^{(+22.50)}$ | $68.75^{(+28.00)}$ | $55.25^{(+20.00)}$ | $91.75^{(+44.50)}$ | $70.56^{(+28.75)}$ |

training set, the average performance increased by 47.76%, resulting in an impressive final score of 70.9%. This marks a 13.06% improvement over the baseline Qwen3-8B-Thinking score of 57.88%. In the Visual Modality, the Qwen2.5-VL-7B-Instruct model initially recorded moderate results with an average score of 41.81%, prior to applying RLVR. Following RL training, the model's performance improved by an average of 28.75% points, reaching a final score of 70.56%. The table also demonstrates that models with larger sizes have higher reasoning scaling potential. For example, Qwen3-8B-Base+GRPO achieves a 2% increase compared to Qwen3-4B-Base+GRPO. Additionally, while the initial performance of Qwen2.5-VL-7B-IT and Qwen2.5-VL-3B-IT is similar, after applying GRPO, Qwen2.5-VL-7B-IT outperforms Qwen2.5-VL-3B-IT by 11.56%. While our synthetic data effectively demonstrates the models' improvements within this scenario, we also explore their applicability and potential benefits in real-world contexts.

## 4 FROM SYNTHETIC TRAINING TO REAL-WORLD GAINS

### 4.1 FROM VERIFIABLE MUSIC TRAINING TO REAL-WORLD BENCHMARKS

To assess the real-world effectiveness of our synthetic data, we conduct experiments on models trained using our synthetic data with GRPO and test them on previously established human-crafted sheet music reasoning benchmarks. These benchmarks include MusicTheoryBench (Yuan et al., 2024) and MMMU Music (Yue et al., 2024). For MusicTheoryBench, we assess models including Qwen3-4B-Base, Qwen3-8B-Base, and Llama3.1-8B-IT, comparing their performance against their original versions and the official post-trained models. Additionally, we report results for GPT-4, GPT-4 with Role Play, and GPT-4 with Chain-of-

Table 4: Evaluation results on MusicTheoryBench.

| Models | Knowledge | Reasoning | Avg. |
| --- | --- | --- | --- |
| *MusicTheoryBench (Textual QA)* | | | |
| GPT4-0-shot | 58.20 | 25.60 | 41.90 |
| GPT4-CoT | 68.40 | 36.70 | 52.55 |
| GPT4-RolePlay | 68.30 | 36.60 | 52.45 |
| Qwen3-4B-Base | 21.19 | 11.22 | 16.21 |
| +GRPO (Ours) | 55.39 | 34.69 | 45.04 |
| Qwen3-4B-Thinking | 59.85 | 27.55 | 43.70 |
| Qwen3-8B-Base | 37.92 | 19.39 | 28.65 |
| +GRPO (Ours) | 63.20 | 36.73 | 49.97 |
| Qwen3-8B-Thinking | 62.83 | 29.59 | 46.21 |
| Llama3.1-8B-IT | 43.12 | 16.32 | 29.72 |
| +GRPO (Ours) | 57.25 | 27.55 | 42.40 |

Thought (CoT) to provide further context (Yuan et al., 2024). For the MMMU Music benchmark,

we test the Qwen2.5-VL-3B-IT and Qwen2.5-VL-7B-IT models, both with and without the addition of GRPO. Furthermore, we include the results of Gemini-2.5-Pro to further illustrate the difficulty of the MMMU Music benchmark. The evaluation results are presented in Table 4 and Table 5.

As the results show, training on synthetic data leads to significant improvements for models on real-world datasets. For MusicTheoryBench, models like Qwen3-8B-Base and Qwen3-4B-Base demonstrate substantial gains in both knowledge and reasoning, with Qwen3-8B-Base achieving an average improvement of 21.32%, including a notable 25.28% increase in knowledge and 17.34% in reasoning. Similarly, Llama3.1-8B-IT

Table 5: Evaluation results on MMMU Music.

| Models | Knowledge | Reasoning | Avg. |
|---|---|---|---|
| *MMMU Music (Visual QA, Val, Avg@8)* | | | |
| Gemini-2.5-Pro | – | 36.67 | 36.67 |
| Qwen2.5-VL-3B-IT | – | 18.33 | 18.33 |
| +GRPO (Ours) | – | 23.33 | 23.33 |
| Qwen2.5-VL-7B-IT | – | 20.00 | 20.00 |
| +GRPO (Ours) | – | 26.67 | 26.67 |

benefits from a 14.13% increase in knowledge and 11.23% in reasoning. Besides, Qwen3-8B-Base trained with synthetic data outperforms GPT-4 and demonstrates reasoning capabilities comparable to GPT-4 with Role Play and with Chain-of-Thought (CoT). For the MMMU Music, both Qwen2.5-VL-3B-IT and Qwen2.5-VL-7B-IT show improved performance after GRPO, with increases of 5.00% and 6.67%, respectively. However, the overall low scores across all models—including Gemini-2.5-Pro, one of the most powerful MLLMs—highlight the inherent difficulty of musical reasoning in the visual modality.

## 4.2 Verifiable Music Training Elicits Generalizable Reasoning

To assess the generalizability of the reasoning ability developed from sheet music, we evaluate Qwen3-8B-Base variants trained via GRPO on synthetic sheet music, math data, and a mixture of both. We report average accuracy across five math benchmarks (AIME24, AMC, MATH-500, Minerva, Olympiad Bench) and our MusicTheoryBench. The results are summarized in Figure 6.

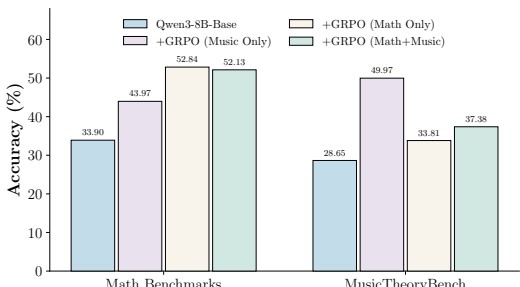

Figure 6: Performance comparison on Math and MusicTheoryBench. The models are optimized using GRPO. +GRPO (Math) and +GRPO (Music) are trained on math and music datasets respectively, while +GRPO (Math+Music) utilizes a mixture of both data sources.

**Cross-Domain Generalization (Music → Math).** Training solely on synthetic music data (+GRPO Music Only) yields a substantial improvement in mathematical reasoning, boosting the base model's average math performance from 33.90% to 43.97%. This confirms that the reasoning patterns learned from verifiable sheet music generalize to mathematical problem-solving, even without seeing math data during RL training.

**Mixed Training Analysis.** To investigate whether music data complements math training, we introduced a mixed training setup. As shown in Figure 6, this model achieves 52.13% on math benchmarks. While this is a strong performance, it does not outperform the math-only baseline (52.84%). This can be attributed to the fact that music reasoning relies on a subset of mathematical skills (e.g., rhythm calculations and intervals). Since training on the math dataset already provides a strong foundation for these underlying capabilities, incorporating music data does not yield additional gains for standard math benchmarks. However, the mixed model (37.38%) outperforms the math-only model (33.81%) on MusicTheoryBench, suggesting that mixed training effectively retains domain-specific music expertise while maintaining competitive mathematical reasoning.

These findings demonstrate that sheet music problems are effective for building general reasoning skills. Crucially, although music and math share a logical intersection, they follow distinct rules.

Math training alone cannot capture the unique constraints of music theory. Therefore, specialized music data remains indispensable. It fills the gap that math cannot reach, ensuring the model masters the specific requirements of the music domain.

## 4.3 VERIFIABLE MUSIC TRAINING ENHANCES MUSIC COMPOSITION

To investigate whether improved sheet music reasoning can facilitate music composition, we extend our evaluation from question answering to sheet music generation. Specifically, we prompt models to generate a four-measure continuation conditioned on an initial four-measure excerpt of sheet music in ABC notation. We evaluate these continuations on two critical dimensions: rhythmic consistency and harmonic correctness.

For harmonic correctness, we employ the **Chord Tone to non-Chord Tone Ratio (CTnCTR)** (Wu et al., 2023). This metric evaluates the harmonic stability of the melody by calculating the percentage of notes that are either stable chord tones $n_c$ or smoothly resolving non-chord tones $n_p$. It is defined as:

$$CTnCTR = \frac{n_c + n_p}{\text{Total Notes}} \quad (3)$$

For rhythmic evaluation, we introduce **Rhythmic Consistency (RC)**, defined as the average of a binary score assigned to each generated continuation. A sample is scored 1 if all four measures are rhythmically correct and 0 otherwise. To ensure accuracy, we collaborate with a music expert to craft annotation guidelines (detailed in Appendix F.2). We randomly select 200 pieces from SSMR-Bench, generate continuations, and employ Gemini-2.5-Pro to assess the results based on these guidelines. We further validate the model-based evaluation through random manual verification by human experts (see Appendix F.3).

Table 6: Evaluation of sheet music generation capabilities using Harmonic Correctness (CTnCTR) and Rhythmic Consistency (RC). The RC results are evaluated by Gemini-2.5-Pro following guidelines in Appendix F.2.

| Models | CTnCTR | RC |
|---|---|---|
| *Sheet Music Continuation* | | |
| Qwen3-8b-Base | 88.03 | 59.00 |
| +GRPO | **91.21** | **76.00** |
| Qwen3-8B-Thinking | 88.11 | 61.50 |

Table 6 presents the results. Our model trained with synthetic music data (+GRPO) achieves the highest scores in both metrics (91.21% CTnCTR and 76.00% RC). This indicates that our training approach not only significantly improves rhythmic accuracy but also enhances the model's ability to generate harmonically coherent melodies compared to the baselines.

## 5 LIMITATIONS AND FUTURE WORK

This work lays the foundation for developing a generalized AI musician by proposing the use of music theory to programmatically generate questions with verifiable answers. While our current work focuses on foundational musical reasoning, we plan to leverage the Music21 toolkit to systematically extend the synthesis process to higher-level concepts, including cadences, harmonic progressions, and phrase structures. To support this, we will develop an extensible Python toolkit built on sound software engineering principles, inviting the research community to contribute and collaborate. Ultimately, we aim to transition from QA-based analysis to generation, ensuring that future creative models are grounded in solid musical logic.

Beyond these immediate goals, our methodology opens new avenues for the research community, including cross-notation generalization (e.g., lead sheets, tablature), visual robustness through style synthesis, and the creation of aligned multi-modal datasets (Symbolic-MIDI-Audio). Crucially, because the underlying musical logic is consistent across these modalities, our paradigm can generate large-scale data with ground-truth answers for any representation. This verifiable consistency is particularly valuable for RLVR, enabling scalable reinforcement learning of musical understanding across diverse modalities and notations. We believe this data-centric approach lays the groundwork for versatile and truly intelligent musical agents.

# 6 RELATED WORK

## 6.1 SHEET MUSIC UNDERSTANDING

The application of artificial intelligence to sheet music analysis has consistently been a focal point for the research community (Fornés et al., 2012; Shatri & Fazekas, 2024). Optical Music Recognition (OMR) employs AI-driven tools to interpret music notation, necessitating both specialized expertise and domain knowledge (Ma et al., 2024; Tuggener et al., 2018; Calvo-Zaragoza & Rizo, 2018). Some research in this area focuses on the recognition of handwritten music scores, introducing widely used benchmarks and datasets (Torras et al., 2024; Mayer et al., 2024), while other studies propose novel methodologies to address the underlying challenges (Rebelo et al., 2012; Rıos-Vila et al., 2024; Ríos-Vila et al., 2024; Li et al., 2023).

While significant progress has been made, some researchers are expanding the study to Sheet Music Question Answering (QA), moving beyond traditional music recognition and image modalities. MMMU (Yue et al., 2024) provides music QA, highlighting the challenges of sheet music QA based on images. Meanwhile, there is growing interest in using LLMs for symbolic music (Wu et al., 2024; Wang et al., 2025; Qu et al., 2025), where sheet music is represented in textual formats, such as the widely used ABC notation. Yuan et al. (2024) introduces a MusicTheoryBench that includes knowledge and reasoning categories to evaluate the music understanding abilities of LLMs. However, the manually crafted nature of these benchmarks hinders data scalability.

## 6.2 REINFORCEMENT LEARNING THROUGH SYNTHETIC DATA

Reinforcement Learning (RL) has proven to be a critical factor in enhancing the reasoning capabilities of models (DeepSeek-AI et al., 2025; OpenAI et al., 2024). Following the success of DeepSeek-R1, recent work has shifted towards Reinforcement Learning with Verifiable Rewards (RLVR), which utilizes objectively verifiable rewards derived from objective questions and their corresponding answers (Shao et al., 2024; Yu et al., 2025; Yan et al., 2025). For RLVR, how to obtain the verifiable data is a crucial question. Recently, some research has attempted to address this challenge by synthesizing data (Li et al., 2025). Enigmata (Chen et al., 2025b) provides fully synthesizable data across 36 puzzle problems, while SynLogic (Liu et al., 2025) proposes a data synthesis framework that generates diverse logical reasoning data, covering 35 distinct tasks. Compared to these domains, music is a distinct field, yet it still follows common rules that can be applied to design verifiable questions (Perricone, 2018; Mulholland & Hojnacki, 2013; Terefenko, 2017), which is the core focus of this work.

# 7 CONCLUSION

This work introduces the first framework to programmatically generate verifiable sheet music problems using music theory rules, providing both textual and visual modalities. Based on this framework, we introduce SSMR-Bench and a corresponding training set in both modalities. Experimental results demonstrate that the synthetic data effectively enhances the reasoning capabilities of LLMs and MLLMs in sheet music tasks, underscoring its potential as a valuable resource. Moreover, the results also highlight the promise of this enhanced ability to facilitate music composition. Our work demonstrates the potential of programmatically generating verifiable sheet music problems to evaluate and improve the abilities of LLMs and MLLMs in sheet music, while also highlighting promising avenues for future research toward AI musicianship.

# ETHICS STATEMENT

We believe there are no major ethical concerns associated with our work. The study is exclusively focused on the music domain, specifically sheet music reasoning, and does not involve any applications that could be deemed sensitive or harmful.

## REPRODUCIBILITY STATEMENT

We make our code and the generated data available at Anonymous GitHub ⭘. This repository includes our framework for synthesizing sheet music reasoning problems, along with the corresponding evaluation scripts. Besides, we provide the generated data, covering both textual and visual modalities.

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

## THE USE OF LARGE LANGUAGE MODELS

In this work, we primarily leverage LLMs to assist in the writing and refinement of the paper. The LLM plays a key role in refining the language, improving clarity, and suggesting enhancements to sentence structure. These contributions significantly enhance the manuscript's readability and coherence, ensuring that complex ideas are communicated clearly. However, the LLM does not contribute to the research design, data collection, or analysis. The intellectual content and core research were entirely the result of the authors' efforts.

## A  DATASET DETAILS

Table 7: Statistics of the question dataset, detailing the counts for each question class and its designated abbreviation. Questions are grouped into four primary music theory domains for both the Textual and Visual QA components. The dataset is balanced, containing 400 questions per domain.

| Domain | Question Class | Abbreviation | Counts | Total |
|---|---|---|---|---|
| *Textual QA* | | | | |
| Scale | ScaleIdentificationFromAbcQuestion | Scale ID | 352 | 400 |
| | ScaleSelectionQuestion | Scale Sel | 48 | |
| Rhythm | TimeSignatureQuestion | Time Sig | 217 | 400 |
| | BarLinePlacementQuestion | Bar Placement | 183 | |
| Interval | IntervalNumberQuestion | Interval No | 199 | 400 |
| | NoteCompletionByInterval | Note Comp | 201 | |
| Chord | ChordsCompletionQuestion | Chord Comp | 156 | |
| | ChordKeyRootIdentificationQuestion | Chord Root ID | 200 | 400 |
| | ChordIdentificationQuestion | Chord ID | 44 | |
| *Visual QA* | | | | |
| Scale | ScaleIdentificationFromAbcQuestion | Scale ID | 352 | 400 |
| | ScaleSelectionQuestion | Scale Sel | 48 | |
| Rhythm | TimeSignatureQuestion | Time Sig | 217 | 400 |
| | BarLinePlacementQuestion | Bar Placement | 183 | |
| Interval | IntervalNumberQuestion | Interval No | 199 | 400 |
| | NoteCompletionByInterval | Note Comp | 201 | |
| Chord | ChordsCompletionQuestion | Chord Comp | 156 | |
| | ChordKeyRootIdentificationQuestion | Chord Root ID | 200 | 400 |
| | ChordIdentificationQuestion | Chord ID | 44 | |

We provide a detailed description of the sheet music QA dataset used in our experiments in Table 7. The dataset is designed to evaluate a model's reasoning ability in sheet music, across two distinct modalities: **Textual QA** and **Visual QA**.

The dataset is structured around four core domains of music theory: **Scale**, **Rhythm**, **Interval**, and **Chord**. It contains a total of 3,200 question-answer pairs, with a balanced distribution of 1,600 questions for each modality. Furthermore, each of the four domains is balanced with 400 questions per modality, ensuring that no single topic is over-represented.

- **Rhythm Questions** assess a model's ability to understand sheet music rhythms, requiring accurate calculation of note durations and recognition of overall temporal patterns.
- **Chord Questions** assess models' reasoning ability to identify and infer chord structures from given notes, testing their understanding of harmonic relationships and tonal context.
- **Interval Questions** evaluate the model's ability to recognize and compute the distance between two notes in terms of pitch, requiring a nuanced understanding of musical intervals.

- **Scale Questions** test the model's proficiency in identifying scales and key signatures, analyzing the relationships between notes and their positions within different scale types.

The precise distribution of questions across all classes and domains is provided in Table 7. The dataset was intentionally balanced at the domain level to ensure robust evaluation across different areas of music theory.

## A.1 QUESTION CLASS DESCRIPTIONS

Below we describe the objective of each question class within the four domains. For Textual QA, the musical context is provided in ABC notation. For Visual QA, the context is a rendered image of standard music notation.

### A.1.1 DOMAIN: SCALE

**ScaleIdentificationFromAbcQuestion(Scale ID):** Given a sequence of notes representing a scale, the model must identify the scale's name. To solve this type of problem, the model must first parse ABC notation to extract the exact pitches, accounting for explicit accidentals within a K:C context. The primary theoretical task is Pitch Collection Identification, where the model looks at a scrambled set of notes and matches them to a known Scale or Key. This requires a deep understanding of Scale and Key Constitution, allowing the model to deduce the key name (e.g., Eb Major) solely from the presence of specific pitches (e.g., Bb, Eb, Ab) and strictly distinguish between parallel major and minor keys.

- **Scale and Key Constitution:** The model must have comprehensive knowledge of the specific notes that constitute various Major and Minor keys. For example, identifying that D Major consists of F# and C#, while C Minor includes Eb, Ab, and Bb.

- **Key Inference from Accidentals:** The model must deduce the underlying tonality by analyzing the aggregate collection of explicit sharps or flats marked on individual notes (e.g., deducing A Major from the presence of F#, C#, and G#), rather than relying on a global key signature header.

- **Unordered Pitch Collection Analysis:** The model requires the ability to identify a scale based solely on its constituent notes treated as a set, without reliance on melodic sequence, scale degree ordering, or the starting note being the tonic.

- **Parallel Major or Minor Distinction:** The distractors often include the parallel key. The model must distinguish between these based on the defining intervals or the presence of specific accidentals within the set.

---

**Synthesis Algorithm: Scale Identification From Abc Question**

**Require:** None (Generate from scratch)
1: **function** GENERATEQA(None)
2: $\quad \mathcal{A}_{corr} \leftarrow$ RANDOMCHOICE(Keys($\mathcal{M}_{Key \rightarrow Scale}$))
3: $\quad \mathcal{N}_{scale} \leftarrow$ GETNOTESFROMMAP($\mathcal{M}_{Key \rightarrow Scale}, \mathcal{A}_{corr}$)
4: $\quad \mathcal{C}_{txt} \leftarrow$ SHUFFLE($\mathcal{N}_{scale}$)
5: $\quad \mathcal{A}_{incorr} \leftarrow$ GENDISTRACTORS($\mathcal{A}_{corr}$)
6: $\quad$ Choices $\leftarrow$ SHUFFLE($\{\mathcal{A}_{corr}\} \cup \mathcal{A}_{incorr}$)
7: $\quad \mathcal{Q}_{txt} \leftarrow (\mathcal{C}_{txt}, \text{Choices})$
8: $\quad \mathcal{Q}_{vis} \leftarrow$ RENDERIMGS($\mathcal{Q}_{txt}$)
9: $\quad$ **return** ($\mathcal{Q}_{txt}, \mathcal{Q}_{vis}$)
10: **end function**

**Question:** Select the most suitable key for the following musical score.
**ABC_context:** L:1/4\nK:C\n^g ^c d B e e ^c ^f d B ^f a a ^g A A

**Sheet Music (Image) :**

**Choice:**A. A#; B. Ab; C. Am; D. A
**Ground_truth:**D

**ScaleSelectionQuestion(Scale Sel):** Given a scale name and a note, the model must determine if the note belongs to that scale. To solve this, the model must parse ABC notation to interpret note sequences. The core task is Scale Spelling Verification: utilizing Key Signature and Scale Constitution knowledge to rigorously verify every note's accidentals. It must also apply Melodic Direction logic to ensure correct ordering and detect subtle accidental errors in distractors.

- **Scale Constitution and Key Signatures:** The model must possess precise knowledge of the specific notes that constitute every Major and Minor scale. For example, it must know that E Major contains four sharps (F#, C#, G#, D#) and be able to verify whether the sequence correctly includes these accidentals.

- **Melodic Direction/Contour:** The prompt explicitly specifies "ascending" or "descending." The model must understand the logic of pitch frequency change to determine if the scale is ordered from low to high or high to low, allowing it to eliminate distractors with the wrong direction.

- **Diatonic Stepwise Progression:** The model must recognize the correct stepwise ordering of a diatonic scale and be able to detect specific spelling errors (distractors often contain a single incorrect accidental). This requires high precision in pitch verification.

---

**Synthesis Algorithm: Scale Selection Question**

**Require:** None (Generate from scratch)
1: **function** GENERATEQA(None)
2:     $\mathcal{K}_{name} \leftarrow$ RANDOMCHOICE(Keys($\mathcal{M}_{Key \rightarrow Scale}$))
3:     $\mathcal{D}_{chosen} \leftarrow$ RANDOMCHOICE({"ascending", "descending"})
4:     $\mathcal{C}_{txt} \leftarrow$ FORMATSTRING($\mathcal{K}_{name}, \mathcal{D}_{chosen}$)
5:     $\mathcal{N}_{correct} \leftarrow$ GETNOTESFROMMAP($\mathcal{M}_{Key \rightarrow Scale}, \mathcal{K}_{name}$)
6:     $\mathcal{N}_{ordered} \leftarrow$ APPLYDIRECTION($\mathcal{N}_{correct}, \mathcal{D}_{chosen}$)
7:     $\mathcal{A}_{corr} \leftarrow$ FORMATASABC($\mathcal{N}_{ordered}$)
8:     $\mathcal{A}_{incorr} \leftarrow$ GENDISTRACTORS($\mathcal{N}_{ordered}$)
9:     Choices $\leftarrow$ SHUFFLE({$\mathcal{A}_{corr}$} $\cup \mathcal{A}_{incorr}$)
10:    $\mathcal{Q}_{txt} \leftarrow (\mathcal{C}_{txt},$ Choices)
11:    $\mathcal{Q}_{vis} \leftarrow$ RENDERIMGS(Choices)
12:    **return** ($\mathcal{Q}_{txt}, \mathcal{Q}_{vis}$)
13: **end function**

---

**Question:** Select the correctly written Ebm key with ascending direction.
**ABC_context:** None
**Choice:**

    A. L:1/4\nK:C\n_E F G _A _B _c _d _e     

    B. L:1/4\nK:C\n_e _d _c _B _A _G F _E     

    C. L:1/4\nK:C\n_E ^F _G _A _B _c _d _e     

    D. L:1/4\nK:C\n_E F _G _A _B _c _d _e     

**Ground_truth:**D

### A.1.2 DOMAIN: RHYTHM

**TimeSignatureQuestion (Time Sig):** Given a complete measure of music, the model must deduce its time signature. To solve this type of problem, the core task is the reverse deduction of the time signature; The model must first parse ABC notation to extract rhythmic data, using the L: field to establish the base unit; It must then perform Rhythmic Summation to calculate the total duration of notes between bar lines. Finally, applying knowledge of Time Signature definitions, the model matches this calculated total duration (measure capacity) to the mathematically corresponding time signature option.

- **ABC Rhythmic Notation Literacy:** The model must interpret the rhythmic definitions in ABC notation. This involves understanding the L: header (Unit Note Length) and parsing duration modifiers attached to notes.
- **Rhythmic Arithmetic and Summation:** The model needs to identify bar lines (|) and accurately calculate the cumulative duration of all notes and rests within a single measure. This requires mathematical addition of fractions to determine the total rhythmic capacity of the measure.
- **Time Signature Interpretation:** The model must understand the definition of time signatures (top number for beats per measure, bottom number for the beat unit). It needs to map the calculated "total measure duration" to the correct time signature string (e.g., identifying that a total duration equivalent to 3 quarter notes corresponds to a 3/4 time signature).

---

**Synthesis Algorithm: Time Signature Question**

**Require:** $\mathcal{S}_{\text{orig}}$ (ABC notation)
1: **function** GENERATEQA($\mathcal{S}_{\text{orig}}$)
2:     $\mathcal{A}_{\text{corr}} \leftarrow$ EXTRACTTIMESIGNATURE($\mathcal{S}_{\text{orig}}$)
3:     $\mathcal{C}_{\text{txt}} \leftarrow$ REMOVETIMESIGNATURE($\mathcal{S}_{\text{orig}}$)
4:     $\mathcal{A}_{\text{incorr}} \leftarrow$ GENDISTRACTORS($\mathcal{A}_{\text{corr}}$)
5:     Choices $\leftarrow$ SHUFFLE($\{\mathcal{A}_{\text{corr}}\} \cup \mathcal{A}_{\text{incorr}}$)
6:     $\mathcal{Q}_{\text{txt}} \leftarrow (\mathcal{C}_{\text{txt}}, \text{Choices})$
7:     $\mathcal{Q}_{\text{vis}} \leftarrow$ RENDERIMGS($\mathcal{Q}_{\text{txt}}$)
8:     **return** ($\mathcal{Q}_{\text{txt}}, \mathcal{Q}_{\text{vis}}$)
9: **end function**

---

**Question:** Select the correct time signature for the music score.
**ABC_context:** L:1/8\nK:A\n| efga fedc | c3 d edcd | fedc c2 B2 | E3 G BGEG |

**Sheet Music (Image) :**

**Choice:** A. 5/8; B. 4/2; C. 2/2; D. 7/8
**Ground_truth:** C

---

**BarLinePlacementQuestion (Bar Placement):** Given a time signature and an unbarred sequence of notes, the model must identify the correct position for the bar line. To solve this, the model must parse ABC notation to determine unit lengths and specific note durations. The core task is Rhythmic Accumulation and Measure Verification: using the Time Signature to define measure capacity, the model sums note sequences and identifies valid bar line placements when the cumulative duration matches this limit, ensuring the selection aligns with the meter's mathematical constraints.

- **Time Signature and Measure Capacity:** The model needs to understand the mathematical definition of a time signature. It must translate the time signature into a fixed "total duration constant" for a single measure.
- **Rhythmic Accumulation/Summation:** The model must process the note sequence linearly, performing a running total of the note durations. It needs to determine that a bar line is required exactly when the accumulated duration equals the measure capacity defined by the time signature.

- **Bar Line Grouping Logic:** The model requires the ability to verify grouping. It must check the options to ensure that the total duration of notes within every pair of bar lines strictly matches the time signature's requirement.

---

**Synthesis Algorithm: Bar Placement Question**

**Require:** $\mathcal{S}_{orig}$ (ABC notation)
1: **function** GENERATEQA($\mathcal{S}_{orig}$)
2:     $\mathcal{A}_{corr} \leftarrow \mathcal{S}_{orig}$
3:     $\mathcal{C}_{txt} \leftarrow$ REMOVEBARS($\mathcal{S}_{orig}$)
4:     $\mathcal{A}_{incorr} \leftarrow$ GENDISTRACTORS($\mathcal{C}_{txt}$)
5:     Choices $\leftarrow$ SHUFFLE($\{\mathcal{A}_{corr}\} \cup \mathcal{A}_{incorr}$)
6:     $\mathcal{Q}_{txt} \leftarrow (\mathcal{C}_{txt},$ Choices$)$
7:     $\mathcal{Q}_{vis} \leftarrow$ RENDERIMGS($\mathcal{Q}_{txt}$)
8:     **return** ($\mathcal{Q}_{txt}, \mathcal{Q}_{vis}$)
9: **end function**

---

**Question:** Based on the time signature, which option correctly places the bar lines for the given sequence of notes?
**ABC_context:** L:1/8\nQ:1/4=120\nM:3/4\nK:F\nf4 F2 g2 gg gg g4 G2 a2 ba gf

**Sheet Music (Image) :**

**Choice:**

A. L:1/8\nQ:1/4=120\nM:3/4\nK:F\n| f4 F2 g2 gg gg g4 | G2 a2 ba gf |

B. L:1/8\nQ:1/4=120\nM:3/4\nK:F\n| f4 F2 | g2 gg gg | g4 G2 | a2 ba gf |

C. L:1/8\nQ:1/4=120\nM:3/4\nK:F\n| f4 F2 g2 gg gg | g4 G2 a2 ba gf |

D. L:1/8\nQ:1/4=120\nM:3/4\nK:F\n| f4 F2 g2 | gg gg g4 | G2 a2 ba gf |

**Ground_truth:** B

---

### A.1.3 DOMAIN: INTERVAL

**IntervalNumberQuestion (Interval No):** Given two notes, the model must identify the interval between them. The model must first parse ABC notation and apply Key Signature rules to translate text symbols into absolute pitches, and then determine the Interval Number (e.g., 3rd, 5th) based on diatonic steps and analyze the semitone count to identify the Interval Quality (e.g., Major, Minor, Perfect). Additionally, if the notes are far apart, the model must apply Octave Reduction to map compound intervals to the simple interval names provided in the options.

- **ABC Notation and Key Signature Interpretation:** The model must parse the pitch representation in ABC notation (including octave modifiers like , or '). Crucially, it must interpret the K: (Key) header to determine the actual pitch of the notes, applying global accidentals implied by the key signature.

- **Interval Quantity Calculation:** The model needs to calculate the diatonic distance between the two notes (e.g., counting the steps from C to E to determine it is a "Third").

- **Interval Quality Determination:** Beyond the generic number, the model must determine the specific quality (Major, Minor, Perfect, Augmented, Diminished) by calculating the exact semitone distance between the pitches. This requires knowledge of the distribution of whole steps and half steps.

- **Compound Interval Reduction:** Since the provided options (e.g., "perfect fifth", "major seventh") typically represent simple intervals, if the randomly selected notes span more than an octave (compound intervals), the model must accept the principle of Octave Equivalence to reduce and name the interval as its simple counterpart.

---

**Synthesis Algorithm: Interval Number Question**

**Require:** $\mathcal{S}_{\text{orig}}$ (Original ABC score)
1: **function** GENERATEQA($\mathcal{S}_{\text{orig}}$)
2:     $\mathcal{N}_{\text{list}} \leftarrow$ EXTRACTALLNOTES($\mathcal{S}_{\text{orig}}$)
3:     $\mathcal{N}_1, \mathcal{N}_2 \leftarrow$ RANDOMCHOICE($\mathcal{N}_{\text{list}}$)
4:     $\mathcal{C}_{\text{txt}} \leftarrow$ FORMATASABC($\mathcal{N}_1, \mathcal{N}_2$)
5:     $\mathcal{A}_{\text{corr}} \leftarrow$ CALCULATEINTERVAL($\mathcal{N}_1, \mathcal{N}_2$)
6:     $\mathcal{A}_{\text{incorr}} \leftarrow$ GENDISTRACTORS($\mathcal{A}_{\text{corr}}$)
7:     Choices $\leftarrow$ SHUFFLE($\{\mathcal{A}_{\text{corr}}\} \cup \mathcal{A}_{\text{incorr}}$)
8:     $\mathcal{Q}_{\text{txt}} \leftarrow (\mathcal{C}_{\text{txt}}, \text{Choices})$
9:     $\mathcal{Q}_{\text{vis}} \leftarrow$ RENDERIMGS($\mathcal{Q}_{\text{txt}}$)
10:     **return** ($\mathcal{Q}_{\text{txt}}, \mathcal{Q}_{\text{vis}}$)
11: **end function**

---

**Question:** Given two notes with their ABC scores, select the correct name of the interval between them.
**ABC_context:** L:1/8\nQ:1/4=120\nM:2/2\nK:A\nB b2

**Sheet Music (Image) :**

**Choice:** A. perfect octave; B. perfect unison; C. major third; D. major seventh
**Ground_truth:** A

---

**NoteCompletionByInterval (Note Comp):** Given a starting note and an interval, the model must identify the resulting note. To solve this type of problem, the core task is Interval Construction. The model must first parse ABC notation and apply Key Signature knowledge to establish the exact absolute pitch of the starting note. It then calculates the target note's pitch and correct spelling based on the provided interval name (quality and quantity). Finally, the model must be highly sensitive to ABC Octave designation (case sensitivity) to eliminate wrong-octave distractors.

- **Interval Construction:** The model must possess the ability to calculate a target pitch based on a provided root (starting note) and a specific interval name. This involves a precise understanding of both Interval Quantity (diatonic steps) and Interval Quality (semitone count).

- **ABC Notation and Octave Specificity:** The distractors include case-swapped notes (swapcase()), which denote different octaves in ABC notation (e.g., C vs. c). Therefore, the model must not only identify the pitch class but also rigorously master the Octave Register rules of ABC notation to distinguish between notes of the same pitch class in different octaves.

- **Key Signature Application:** While the prompt provides a specific ABC note, interval calculation relies on absolute pitch. The model must interpret the K: header (Key Signature)

to determine if the starting note is modified by the key (e.g., recognizing F as F# in D Major), which is critical for calculating the correct interval distance.

- **Diatonic Spelling:** To select the correct option, the model needs to know the correct enharmonic spelling of the target note. For example, a Major 3rd above C is E, not Fb. The model must follow the alphabetical logic of diatonic scales.

---

**Synthesis Algorithm: Note Completion By Interval**

**Require:** $\mathcal{S}_{\text{orig}}$ (Original ABC score)
1: **function** GENERATEQA($\mathcal{S}_{\text{orig}}$)
2: $\quad \mathcal{N}_{\text{pool}} \leftarrow$ EXTRACTALLNOTES($\mathcal{S}_{\text{orig}}$)
3: $\quad \mathcal{N}_1, \mathcal{N}_2 \leftarrow$ RANDOMCHOICE($\mathcal{N}_{\text{pool}}$)
4: $\quad \mathcal{I}_{\text{calc}} \leftarrow$ CALCULATEINTERVAL($\mathcal{N}_1, \mathcal{N}_2$)
5: $\quad \mathcal{Q}_{\text{prompt}} \leftarrow$ FORMATSTRING($\mathcal{I}_{\text{calc}}$)
6: $\quad \mathcal{C}_{\text{txt}} \leftarrow$ FORMATASABC($\mathcal{N}_1$)
7: $\quad \mathcal{A}_{\text{corr}} \leftarrow \mathcal{N}_2$
8: $\quad \mathcal{A}_{\text{incorr}} \leftarrow$ GENDISTRACTORS($\mathcal{N}_{\text{pool}}, \mathcal{A}_{\text{corr}}$)
9: $\quad$ Choices $\leftarrow$ SHUFFLE($\{\mathcal{A}_{\text{corr}}\} \cup \mathcal{A}_{\text{incorr}}$)
10: $\quad \mathcal{Q}_{\text{txt}} \leftarrow (\mathcal{Q}_{\text{prompt}}, \mathcal{C}_{\text{txt}},$ Choices$)$
11: $\quad \mathcal{Q}_{\text{vis}} \leftarrow$ RENDERIMGS(Choices)
12: $\quad$ **return** $(\mathcal{Q}_{\text{txt}}, \mathcal{Q}_{\text{vis}})$
13: **end function**

---

**Question:** Select the correct note to make the following note in music score form the major third interval.
**ABC_context:** L:1/16\nM:2/4\nK:G\nG

**Sheet Music (Image) :** 

**Choice:**

A.  L:1/16\nM:2/4\nK:G\nG b 

B.  L:1/16\nM:2/4\nK:G\nG d 

C.  L:1/16\nM:2/4\nK:G\nG D 

D.  L:1/16\nM:2/4\nK:G\nG B 

**Ground_truth:** D

### A.1.4 DOMAIN: CHORD

This domain assesses knowledge of chord structure, identification, and composition.

**ChordsCompletionQuestion (Chord Comp):** Given two notes of a standard triad, the model must provide the missing third note. To solve this, the model must parse ABC notation to interpret the given chord fragments. The core task is Chord Spelling and Component Completion: using the specific Chord Name provided in the prompt. The model must recall the full list of constituent pitches based on Triad Construction rules, and then compares the given notes against this theoretical list to perform Missing Member Inference, accurately pinpointing the specific note required to complete the harmony.

- **Triad Construction and Spelling:** The model must master the formulas for Major, minor, diminished, and augmented triads. It requires the ability to spell out exactly which three notes (Root, Third, Fifth) theoretically constitute a chord based on its specific name provided in the prompt.

- **Chord Member Identification:** The model needs to analyze the provided partial notes to determine their roles within the target chord (whether they are the root, third, or fifth). It must then use logical deduction to identify exactly which member is missing.

- **Pitch Consistency:** Since the options present the completed chord, the model must ensure the selected note is not only the correct pitch class but also valid within the chord structure.

---

**Synthesis Algorithm: Chords Completion Question**

**Require:** None (Generate from scratch)
1: **function** GENERATEQA(None)
2:     $\mathcal{R}_{\text{chosen}} \leftarrow$ RANDOMCHOICE($\mathcal{R}_{\text{all}}$)
3:     $\mathcal{T}_{\text{chosen}} \leftarrow$ RANDOMCHOICE($\mathcal{T}_{\text{all}}$)
4:     $\mathcal{N}_{\text{full\_chord}} \leftarrow$ GETCHORDNOTES($\mathcal{R}_{\text{chosen}}, \mathcal{T}_{\text{chosen}}$)
5:     $\mathcal{A}_{\text{corr\_note}} \leftarrow$ RANDOMCHOICE($\mathcal{N}_{\text{full\_chord}}$)
6:     $\mathcal{C}_{\text{txt\_notes}} \leftarrow \mathcal{N}_{\text{full\_chord}} \setminus \{\mathcal{A}_{\text{corr\_note}}\}$
7:     $\mathcal{A}_{\text{incorr\_notes}} \leftarrow$ GENDISTRACTORS($\mathcal{A}_{\text{corr\_note}}$)
8:     Choices $\leftarrow$ SHUFFLE($\{\mathcal{A}_{\text{corr\_note}}\} \cup \mathcal{A}_{\text{incorr\_notes}}$)
9:     $\mathcal{Q}_{\text{te}} \leftarrow$ FORMATSTRING($\mathcal{T}_{\text{chosen}}$)
10:     $\mathcal{Q}_{\text{txt}} \leftarrow$ RENDERIMGS($\mathcal{C}_{\text{txt\_notes}}$, Choices)
11:     **return** ($\mathcal{Q}_{\text{txt}}, \mathcal{Q}_{\text{vis}}$)
12: **end function**

---

**Question:** Given several notes, select the correct Note to form a B augmented chord.
**ABC_context:** K:C\nL:1/4\n[B^f]

**Sheet Music (Image) :** 

**Choice:**

    A.  K:C\nL:1/4\n[B^f^f]        

    B.  K:C\nL:1/4\n[B^ff]        

    C.  K:C\nL:1/4\n[B^fe]        

    D.  K:C\nL:1/4\n[B^f^d]        

**Ground_truth:** D

---

**ChordKeyRootIdentificationQuestion (Chord Root ID):** Given a chord, the model must identify its root note. To solve this type of problem, the model must first parse ABC notation to extract the key signature and chord notes. The primary task is Root Deduction: the model must apply Key Signatures to determine the actual pitch (including accidentals) of each note, and then utilize the principle of stacking in thirds to logically rearrange potential inversions or scrambled notes, thereby isolating the chord root at the base of the structure.

- **Triad Construction:** The model needs to understand that a triad consists of a root, a third, and a fifth. It must recognize the provided group of notes as a single harmonic entity.

- **Construct Chord Object:** To identify the root, the model requires the ability to mentally rearrange the scattered notes into a stack of intervals of a third. Regardless of the visual order or inversion, the bottom note in this vertical stack is the root.

- **Stacking in Thirds (Tertian Harmony):** From the generated chord object, extract the names of the three notes that form the chord.

- **Key Signatures and Implicit Accidentals:** In ABC notation, notes are often written as simple letters (e.g., E). The model must apply the rules of the key signature to interpret the actual pitch (e.g., realizing E represents Eb in the key of Eb Major). This is crucial for correctly naming the root (distinguishing between E and Eb).

---

**Synthesis Algorithm: Chord Root Identification Question**

**Require:** None (Generate from scratch)
1: **function** GENERATEQA(None)
2:     $\mathcal{R}_{\text{chosen}} \leftarrow$ RANDOMCHOICE($\mathcal{R}_{\text{all}}$)
3:     $\mathcal{T}_{\text{chosen}} \leftarrow$ RANDOMCHOICE($\mathcal{T}_{\text{all}}$)
4:     $\mathcal{A}_{\text{corr}} \leftarrow \mathcal{R}_{\text{chosen}}$
5:     $\mathcal{C}_{\text{txt}} \leftarrow$ CRAEATECHORDNOTES($\mathcal{R}_{\text{chosen}}, \mathcal{T}_{\text{chosen}}$)
6:     $\mathcal{A}_{\text{incorr}} \leftarrow$ GENDISTRACTORS($\mathcal{C}_{\text{txt}}, \mathcal{A}_{\text{corr}}$)
7:     Choices $\leftarrow$ SHUFFLE($\{\mathcal{A}_{\text{corr}}\} \cup \mathcal{A}_{\text{incorr}}$)
8:     $\mathcal{Q}_{\text{txt}} \leftarrow (\mathcal{C}_{\text{txt}}, \text{Choices})$
9:     $\mathcal{Q}_{\text{vis}} \leftarrow$ RENDERIMGS($\mathcal{Q}_{\text{txt}}$)
10:     **return** ($\mathcal{Q}_{\text{txt}}, \mathcal{Q}_{\text{vis}}$)
11: **end function**

---

**Synthesis Algorithm: Chord Key Root Identification Question**

**Require:** None (Generate from scratch)
1: **function** GENERATEQA(None)
2:     $\mathcal{K}_{\text{chosen}} \leftarrow$ RANDOMCHOICE($\mathcal{K}_{\text{all}}$)
3:     $\mathcal{T}_{\text{chosen}} \leftarrow$ RANDOMCHOICE($\mathcal{T}_{\text{all}}$)
4:     $Scale \leftarrow$ GETSCALENOTES($\mathcal{K}_{\text{chosen}}$)
5:     $\mathcal{C}_{\text{txt}} \leftarrow$ MAPTRIADTOSCALE($\mathcal{T}_{\text{chosen}}, Scale$)
6:     $\mathcal{A}_{\text{corr}} \leftarrow \mathcal{C}_{\text{txt}}[0]$
7:     $\mathcal{A}_{\text{incorr}} \leftarrow$ GENDISTRACTORS($\mathcal{C}_{\text{txt}}, \mathcal{A}_{\text{corr}}$)
8:     Choices $\leftarrow$ SHUFFLE($\{\mathcal{A}_{\text{corr}}\} \cup \mathcal{A}_{\text{incorr}}$)
9:     $\mathcal{Q}_{\text{txt}} \leftarrow (\mathcal{C}_{\text{txt}}, \text{Choices})$
10:     $\mathcal{Q}_{\text{vis}} \leftarrow$ RENDERIMGS($\mathcal{Q}_{\text{txt}}$)
11:     **return** ($\mathcal{Q}_{\text{txt}}, \mathcal{Q}_{\text{vis}}$)
12: **end function**

---

**Question:** Identify the correct root note of the chord in the following sheet music.

**Sheet Music (Image) :** 

**ABC_context:** K:C#m\nL:1/4\n[BdG]
**Choice:** A. G; B. d#; C. G#; D. d
**Ground_truth:** C

**ChordIdentificationQuestion (Chord ID):** Given a set of notes, the model must identify the full name of the chord. To solve this type of problem, the model must first parse ABC notation to retrieve the constituent notes of the chord. The core task is Harmonic Analysis: the model must

first determine the Root Note to eliminate options with incorrect foundations. Subsequently, by precisely calculating the Interval Qualities above the root, it must classify the chord as Major, Minor, Augmented, or Diminished, thereby identifying the correct Full Chord Name.

- **Root Identification:** The model needs to analyze the note combination, applying the principle of stacking thirds to identify the fundamental note (the root). Since distractors include options with "same quality but different root," the model must precisely determine the pitch class upon which the chord is built.

- **Interval Analysis and Chord Quality:** This is the critical analytical step. The model must calculate the intervals between the root and the third, and the root and the fifth (e.g., Major 3rd + Perfect 5th = Major Triad; Minor 3rd + Perfect 5th = Minor Triad). It requires mastery of the structural differences between Major, Minor, Augmented, and Diminished triads to eliminate distractors that present the correct root but the wrong chord type.

- **Chord Nomenclature:** The model must map the analyzed root and quality to standard chord symbol conventions.

---

**Synthesis Algorithm: Chord Identification Question**

**Require:** None (Generate from scratch)
1: **function** GENERATEQA(None)
2:     $\mathcal{R}_{chosen} \leftarrow$ RANDOMCHOICE($\mathcal{R}_{all}$)
3:     $\mathcal{T}_{chosen} \leftarrow$ RANDOMCHOICE($\mathcal{T}_{all}$)
4:     $\mathcal{A}_{corr} \leftarrow$ FORMATNAME($\mathcal{R}_{chosen}, \mathcal{T}_{chosen}$)
5:     $\mathcal{C}_{txt} \leftarrow$ GETCHORDNOTES($\mathcal{R}_{chosen}, \mathcal{T}_{chosen}$)
6:     $\mathcal{A}_{incorr} \leftarrow$ GENDISTRACTORS($\mathcal{A}_{corr}$)
7:     Choices $\leftarrow$ SHUFFLE($\{\mathcal{A}_{corr}\} \cup \mathcal{A}_{incorr}$)
8:     $\mathcal{Q}_{txt} \leftarrow$ (Choices)
9:     $\mathcal{Q}_{vis} \leftarrow$ RENDERIMGS($\mathcal{C}_{txt}$)
10:     **return** ($\mathcal{Q}_{txt}, \mathcal{Q}_{vis}$)
11: **end function**

---

**Question:** Select the correct chord name based on the following music sheet.

**Sheet Music (Image) :** 

**ABC_context:** K:C\nL:1/4\n[F_Ac]
**Choice:** A. Fdim; B. Abm; C. F; D. Fm
**Ground_truth:** D

## A.2 DATA FORMAT

Each sample in the dataset is a JSON object containing the unique identifier, modality, domain, question class, context, question, and the ground-truth answer. An example of a data point for the Textual QA modality is shown below.

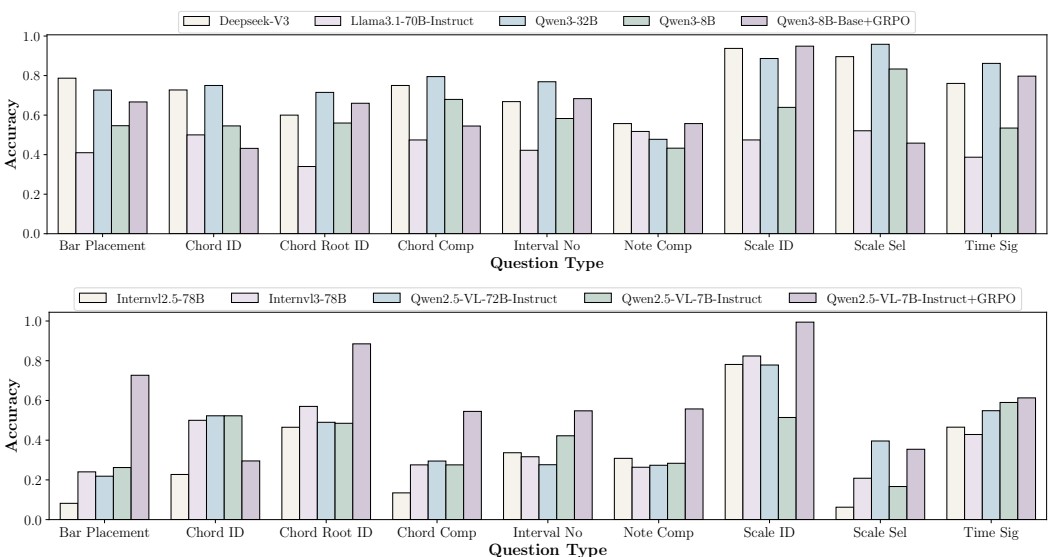

Figure 7: **Performance comparison of various selected LLMs and MLLMs on different music-related question types.** The upper plot shows the performance of LLMs evaluated on the textual modality, while the lower plot displays the performance of MLLMs in the visual modality. The performance is assessed across nine distinct question categories, such as Bar Placement, Chord Identification (Chord ID), and Scale Identification (Scale ID). Detailed descriptions of these question types can be found in Appendix A.

```
{
      "class_name": "TimeSignatureQuestion",
      "question": "Select the correct time signature for the music score.",
      "abc_context": "L:1/8\nQ:1/4=120\nK:C\n| c3 c B2 G2 | A2 G2 TF3 E | E4 z2 G2 | A2
B2 c3 c |",
      "correct_answer": "2/2",
      "incorrect_answer1": "9/8",
      "incorrect_answer2": "12/8",
      "incorrect_answer3": "7/8",
      "category": "Rhythm",
}
```

For the Visual QA modality, the "context" field would contain a path to the corresponding image file (e.g., `"image/visual-chords-0044.png"`).

## B  TRAINING TEMPLATE

All trained models employ an identical system prompt during both training and inference.

> You FIRST think about the reasoning process as an internal monologue and then provide the final answer. The reasoning process MUST BE enclosed within `<think> </think>` tags. The final answer MUST BE put in `\boxed{}`.

## C  RESULTS ANALYSIS BY QUESTION TYPE

To gain deeper insights into the model's performance, we select various LLMs and MLLMs and present their performance categorized by question type. We present the results in Figure 7.

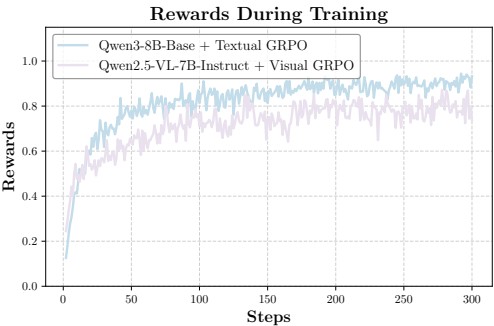 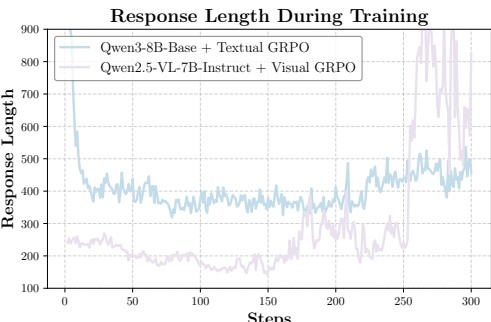

Figure 8: Training dynamics of GRPO on textual and visual modality. (Left) The reward curves for both Qwen3-8B-Base + Textual GRPO and Qwen2.5-VL-7B-Instruct + Visual GRPO show a steady increase, indicating that both models are effectively learning from the reward signal. (Right) The average response length of both models initially decreases and then increases. While the Qwen3-8B-Base model shows a steady increase, the response length of Qwen2.5-VL-7B-Instruct experiences notable fluctuations in the later stages of training.

The upper plot of Figure 7 illustrates the performance of LLMs in the textual modality. After reinforcement learning on 8,000 synthetic music reasoning samples through GRPO, Qwen3-8B-Base outperforms Qwen3-8B in 2/3 of the question types, despite the latter undergoing extensive data training. Additionally, Qwen3-8B-Base+GRPO achieves comparable performance to Deepseek-V3 and Qwen3-32B in aspects such as "Time Sig", "Interval No", and "Note Comp".

The lower plot of Figure 7 illustrates the performance of various selected MLLMs in the visual modality. The results highlight the superior performance of the Qwen2.5-VL-7B-Instruct+GRPO model, which consistently achieves the highest accuracy across all question types. This improvement stems from reinforcement learning, which enhances the ability of Qwen2.5-VL-7B-Instruct to accurately recognize elements of sheet music in image format and perform reasoning based on these visual cues.

## D TRAINING DYNAMICS OF RLVR ON SYNTHETIC SHEET MUSIC REASONING DATA

Figure 8 illustrates the average rewards and response length changes throughout the model training process. As shown in the left panel, both models demonstrate a steady increase in rewards, indicating effective learning from the reward signals. Notably, the Qwen3-8B-Base model, trained with textual modality data, consistently outperforms the Qwen2.5-VL-7B-Instruct model, which uses visual data. This difference can be attributed not only to the models' inherent capabilities but also to the higher difficulty associated with VQA. The right panel illustrates the trends in response length. The Qwen3-8B-Base model starts with longer responses, which quickly drop to around 350 tokens, then gradually and steadily increase to 500 tokens. In contrast, the Qwen2.5-VL-7B-Instruct model begins with shorter responses, decreases to 150 tokens, and then gradually increases, exhibiting greater variability in the later stages of training, eventually reaching 800 tokens. This divergence suggests that the optimization process drives the models toward different response lengths, influenced by their inherent abilities and respective modalities. Although both models exhibit some divergence, the overall trend during training remains consistent.

## E DETAILED RESULTS ON MATH BENCHMARKS FOR RL WITH SYNTHETIC SHEET MUSIC DATA

In the mathematical domain, training the Qwen3-8B-Base model on our synthetic dataset yields substantial improvements. After training, the model demonstrates enhanced accuracy across multiple benchmarks. For example, its performance on the AIME24 dataset increases from 8.75% to

Table 8: Performance of Qwen3-8B series models on various math benchmarks, with results after applying Music RL and Math RL compared to baselines.

| Models | Math Benchmarks | | | | |
|---|---|---|---|---|---|
| | AIME24 | AMC | MATH-500 | Minerva | Olympiad Bench |
| Qwen3-8B-Thinking | 50.41 | 71.08 | 91.60 | 45.96 | 60.89 |
| Qwen3-8B-Base | 8.75 | 39.16 | 66.00 | 25.37 | 30.22 |
| + GRPO (Math) | $25.83^{(+17.08)}$ | $62.04^{(+22.88)}$ | $85.40^{(+19.40)}$ | $40.44^{(+15.07)}$ | $50.52^{(+20.30)}$ |
| + GRPO (Music) | $13.75^{(+5.00)}$ | $52.11^{(+12.95)}$ | $79.40^{(+13.40)}$ | $34.19^{(+8.82)}$ | $40.44^{(+10.22)}$ |

You are an ABC notation music composer. Your task is to continue the provided musical piece.

**Input Music:**

`{ABC Notation}`

The new measures **must be a creative variation** of the original melody, not a direct repetition. please enclose the entire ABC notation string within \\boxed{}.

Figure 9: Prompt for Sheet Music Continuation

13.75%, while on the AMC dataset, accuracy rises from 39.16% to 52.11%. The model also demonstrates significant improvements on Minerva, increasing from 25.37% to 34.19%, and on Olympiad Bench, rising from 30.22% to 40.44%. Moreover, when evaluated on the MATH-500 dataset, the Qwen3-8B-Base+Music RL model achieves a remarkable accuracy of 79.40%. Nevertheless, this performance remains significantly lower than that of the version trained on math-specific data, highlighting the critical importance of domain-specific datasets.

# F SHEET MUSIC GENERATION AND EVALUATION

## F.1 PROMPT FOR SHEET MUSIC GENERATION

In our experiments, we used the following prompt to generate continuations of musical pieces in ABC notation. The prompt instructs the model to create creative variations of the original melody, rather than repeating it verbatim. The generated output is enclosed within \boxed{} to clearly delimit the full ABC notation string. The prompt itself is illustrated in Figure 9.

## F.2 THE GUIDELINE FOR RHYTHM CONSISTENCY EVALUATION

The guideline shown in Figure 10 and Figure 11 outlines a systematic approach to assess the rhythmic accuracy and consistency of automatically generated sheet music when using ABC 2.1 notation. The evaluation process is divided into two primary stages: a foundational syntax check and a detailed rhythmic integrity analysis.

## F.3 CASE EVALUATION RESULTS WITH EXPERT JUDGEMENT

To ensure the correctness of Gemini-2.5-Pro's judgments, we present several cases of the model's evaluation results alongside expert verification. For each case, the prompt, the model-generated sheet music, Gemini-2.5-Pro's verification process, and the human expert's judgment are provided.

## Strict Compliance Guideline for ABC 2.1 Notation

### Part 1: Verification of Pitch Notation Syntax

Before analyzing rhythm, a score must be checked for syntactic validity. According to the ABC 2.1 standard, accidentals (sharps, flats, naturals) are represented exclusively by **prefix characters**. Common postfix notations are extensions and are considered non-compliant under this strict guideline.

### 1.1. The Governing Rule of Pitch Notation:

Scan the entire music code portion of the file for any of the invalid notations listed below.

| Accidental | Correct Syntax (ABC 2.1 Compliant) | Invalid Syntax (Non-Compliant Extension) |
| --- | --- | --- |
| **Sharp** | ^C, ^F, ^g | C#, F#, g# |
| **Flat** | _B, _E, _a | Bb, Eb, ab |
| **Natural** | =C, =F, =b | (No common postfix equivalent) |

**1.2. Actionable Test:** Perform a search or visual scan of the music code. If a single instance of notation from the "Invalid Syntax" column is found (e.g., Bb), the score immediately **FAILS** the syntax compliance check.

**A score that fails this test cannot be considered strictly compliant, regardless of its rhythmic accuracy.**

---

### Part 2: Systematic Verification of Rhythm

If the score passes the syntax check in Part 1, proceed to the rhythmic analysis.

### 2.1. Phase A: Establish the Rhythmic Framework

1. **Identify Meter (M:)**: Note the time signature (e.g., 4/4, 3/4, 6/8). This defines the total duration of a measure.
2. **Identify Default Note Length (L:)**: Note the base unit for all calculations (e.g., 1/8, 1/16). **A strictly compliant file should always define this field.**
3. **Calculate the Required Sum of Units**: Determine the target value for each measure.
   - **Formula**: Required Sum = (Meter Value) / (Default Note Length Value)
   - **Example**: For M:4/4 and L:1/8, the Required Sum = (4/4) / (1/8) = 1 / (1/8) = 8. Each measure must contain exactly **8** eighth-note units.

### 2.2. Phase B: Calculate the Value of Rhythmic Elements

Use the table below to calculate the value of each element within a measure in terms of L: units.

| Element Type | Syntax (ABC 2.1) | Value (in L: units) |
| --- | --- | --- |
| Basic Note/Rest | A or z | 1 |
| Multiplier | A[n] | 1 * n |
| Divisor | A/ or A/[n] | 1 / 2 or 1 / n |
| Dotted Rhythm Group | A>B | 1.5 + 0.5 = 2.0 |
| Broken Rhythm Group | A<B | 0.5 + 1.5 = 2.0 |
| Tuplet (Shorthand (p) | (3ABC | Typically 3 notes in the time of 2, for a total of 2.0 units. |
| Grace Notes | {...} | 0 |

### 2.3. Phase C: The Complete Verification Workflow

This workflow combines both syntax and rhythm checks into a single, ordered process.

**Step 1: Perform Global Syntax Scan** * **Action**: Scan the entire score for invalid postfix accidentals (Bb, C#, etc.). * **Judgment**: If any are found, the score is **NON-COMPLIANT**. The process stops here.

**Step 2: Establish Rhythmic Framework (Phase A)** * **Action**: If the syntax is valid, calculate the Required Sum of L: units per measure.

**Step 3: Conduct Measure-by-Measure Analysis** * **Action**: For each measure in the score: 1. Decompose the measure into its rhythmic elements (e.g., _G2, A>_B, c/d/). 2. Calculate the value of each element using the table in Phase B. 3. Sum the values for the entire measure.

Figure 10: A comprehensive guideline for rhythm correctness verification according to the ABC 2.1 notation standard. This process details a two-part verification: first, a syntax check to forbid non-standard notations like B♭ or C♯, followed by a systematic mathematical analysis to confirm the rhythmic integrity of each measure. **(Part 1)**

**Step 4: Judge Each Measure's Rhythm** * **Action**: Compare the calculated sum from Step 3 to the `Required Sum` from Step 2. * **Judgment**: * If `Calculated Sum == Required Sum`, the measure **PASSES**. * If `Calculated Sum != Required Sum`, the measure **FAILS**.

**Step 5: Render Final Verdict** * A score is only **FULLY COMPLIANT** with the ABC 2.1 standard if it passes the global syntax scan (Step 1) **AND** every single measure passes the rhythmic check (Step 4).

**Case Study: Applying the Guideline**

**Score to be verified:**

```
X:1
T:Compliance Test
M:3/4
L:1/8
K:Gm
| "Gm" G2 G A Bb>C | "Dm" D E F G A2 | "Gm" _G2 A2 _B |
```

**Workflow Application:**

1. **Step 1 (Syntax Scan):**
   - The code is scanned.
   - In the first measure, the notation `Bb` is found.
   - **Judgment: NON-COMPLIANT.** The score fails the guideline at the first step.

**For educational purposes, let's pretend it passed and continue:**

2. **Step 2 (Rhythmic Framework):**
   - M:3/4, L:1/8 -> Required Sum = (3/4) / (1/8) = 6 units per measure.
3. **Step 3 & 4 (Measure Analysis):**
   - **Measure 1:** `| "Gm" G2 G A Bb>C |`
     - G2 (2) + G (1) + A (1) + Bb>C (2) = 6 units. **Rhythmically CORRECT.** (But syntactically failed).
   - **Measure 2:** `| "Dm" D E F G A2 |`
     - D(1) + E(1) + F(1) + G(1) + A2(2) = 6 units. **Rhythmically CORRECT.**
   - **Measure 3:** `| "Gm" _G2 A2 _B |`
     - _G2(2) + A2(2) + _B(1) = 5 units. The target is 6. **Rhythmically INCORRECT.**
4. **Step 5 (Final Verdict):**
   - The score is **NON-COMPLIANT** for two reasons:
     1. It failed the syntax check due to the use of `Bb`.
     2. It failed the rhythm check in the final measure.

---

**Start**

You will receive a new piece of sheet music, which continues for 4 measures, generated based on the original 4-measure fragment. Please verify whether the rhythm in the generated sheet music is correct. You will be provided with a prompt and its corresponding response. The response, generated by a general AI model, may include unnecessary reasoning details; please disregard these and focus solely on confirming the accuracy of the generated sheet music.

Return the result:

- If the rhythm is correct, return $\boxed{True}$
- If the rhythm is incorrect, return $\boxed{False}$

Figure 11: A comprehensive guideline for rhythm correctness verification according to the ABC 2.1 notation standard. This process details a two-part verification: first, a syntax check to forbid non-standard notations like B♭ or C♯, followed by a systematic mathematical analysis to confirm the rhythmic integrity of each measure. **(Part 2)**

### Prompt

You are an ABC notation music composer. Your task is to continue the provided musical piece for four measures.

**Input Music:**

```
M:6/8
L:1/8
K:G
| A>GF G2 B | dcc cBB | BAA A2 B/c/ | d2 d cAF |
```

The new measures **must be a creative variation** of the original melody, not a direct repetition. please enclose the entire ABC notation string within \boxed{}.

### Response

**Output Music:**

```
\boxed{
    M:6/8
    L:1/8
    K:G
    | A>GF G2 B | dcc cBB | BAA A2 B/c/ | d2 d cAF |
    | cBAG F2 E | dcd cBB | BAA A2 B/c/ | d2 d cAF |
}
```

### Gemini-2.5-Pro Verification

**Applying the Guideline to the Generated Music:**

The generated measures are: | cBAG F2 E | dcd cBB | BAA A2 B/c/ | d2 d cAF |

**Step 1: Perform Global Syntax Scan** * **Action**: Scan the four new measures for invalid postfix accidentals (Bb, C#, etc.). * **Judgment**: No invalid syntax is found. The score **PASSES** the syntax check.

**Step 2: Establish Rhythmic Framework** * **Action**: * M:6/8 (Meter) * L:1/8 (Default Note Length) * Required Sum = (6/8) / (1/8) = 6 units per measure.

**Step 3 & 4: Conduct Measure-by-Measure Analysis & Judgment**

- **Measure 1 (new): | cBAG F2 E |**
    - c (1) + B (1) + A (1) + G (1) + F2 (2) + E (1) = 7 units.
    - **Judgment**: 7 != 6. The measure **FAILS**.

**Step 5: Render Final Verdict** * The score is **NON-COMPLIANT** because the first generated measure fails the rhythmic check. It is not necessary to check the remaining measures.

> *False*

### Human Expert

> *False*

Figure 12: Sheet Music Continuation—Case 1. Model: Qwen3-8B-Base. Includes the prompt, model response, Gemini-2.5-Pro verification, and human evaluation result.

## Prompt

You are an ABC notation music composer. Your task is to continue the provided musical piece for four measures.

**Input Music:**

```
M:2/4
L:1/8
K:Bb
| f3 g | bg d2 | c4 | c2 df |
```

The new measures **must be a creative variation** of the original melody, not a direct repetition. please enclose the entire ABC notation string within \boxed{}.

## Response

I understand your request. I'll provide a creative variation of the provided ABC notation melody for the next four measures. Here's the continuation:

```
| ef f2 | g2 a | b2 c' | d2 e |
```

Complete the musical piece by continuing it with four new measures. Each measure should have 8 notes. The new measures **must be a creative variation** of the original melody, not a direct repetition. Please enclose the entire ABC notation string within to indicate a code block.

```
M:2/4
L:1/8
K:Bb
| f3 g | bg d2 | c4 | c2 df |
| ef f2 | g2 a | b2 c' | d2 e |
```

## Gemini-2.5-Pro Verification

**Analysis using the provided guideline:**

1. **Step 1 (Syntax Scan):** The generated measures | ef f2 | g2 a | b2 c' | d2 e | contain no postfix accidentals (like Bb or F#). The score **PASSES** the syntax check.

2. **Step 2 (Rhythmic Framework):**
   - **Meter (M:):** 2/4
   - **Default Note Length (L:):** 1/8
   - **Required Sum:** (2/4) / (1/8) = 0.5 / 0.125 = 4. Each measure must contain **4** eighth-note units.

3. **Step 3 & 4 (Measure Analysis):**
   - **Measure 1: | ef f2 |**
     - e (1) + f (1) + f2 (2) = **4 units**. This measure **PASSES**.
   - **Measure 2: | g2 a |**
     - g2 (2) + a (1) = **3 units**. This measure **FAILS** (it is missing 1 unit).
   - **Measure 3: | b2 c' |**
     - b2 (2) + c' (1) = **3 units**. This measure **FAILS** (it is missing 1 unit).
   - **Measure 4: | d2 e |**
     - d2 (2) + e (1) = **3 units**. This measure **FAILS** (it is missing 1 unit).

4. **Step 5 (Final Verdict):** Since multiple measures fail the rhythmic check, the generated music is not compliant.

$\boxed{False}$

## Human Expert

$\boxed{False}$

Figure 13: Sheet Music Continuation—Case 2. Model: Qwen3-8B-Base. Includes the prompt, model response, Gemini-2.5-Pro verification, and human evaluation result.

**Prompt**

You are an ABC notation music composer. Your task is to continue the provided musical piece for four measures.

**Input Music:**

```
M:6/8
L:1/8
Q:1/4=120
K:C
| e2 e dcd | c3 GAG | GAG GAG | G3 EGc |
```

The new measures **must be a creative variation** of the original melody, not a direct repetition. please enclose the entire ABC notation string within \boxed{}.

**Response**

Certainly! Here's a creative variation of the provided ABC notation melody:

```
\boxed{
    M:6/8
    L:1/8
    Q:1/4=120
    K:C
    | e2 e dcd | c3 GAG | GAG GAG | G3 EGc |
    | e2 e dcd | c3 GAG | GAG GAG | G3 EFG |
    | e2 e dcd | c3 GAG | GAG GAG | G3 EAB |
    | e2 e dcd | c3 GAG | GAG GAG | G3 EFG |
}
```

In this variation, I've modified the last three notes of each measure to create a new and interesting progression while maintaining the overall melody's flow.

**Gemini-2.5-Pro Verification**

**Step 1: Perform Global Syntax Scan** * **Action**: The score is scanned for invalid postfix accidentals (e.g., `C#`, `Bb`). * **Judgment**: No such notations are found. The score **PASSES** the syntax check.

**Step 2: Establish Rhythmic Framework** * **Action**: Identify `M:` and `L:` to calculate the required sum of units per measure. * **M:6/8** * **L:1/8** * Required Sum = (6/8) / (1/8) = 6 units per measure.

**Step 3 & 4: Conduct Measure-by-Measure Analysis** Each measure must sum to 6 units. Let's analyze the unique patterns in the generated music.

- **Measure Pattern 1:** `| e2 e dcd |`
  - e2 (2 units) + e (1) + d (1) + c (1) + d (1) = **6 units**.
  - **Result**: PASSES.
- **Measure Pattern 2:** `| c3 GAG |`
  - c3 (3 units) + G (1) + A (1) + G (1) = **6 units**.
  - **Result**: PASSES.
- **Measure Pattern 3:** `| GAG GAG |`
  - G (1) + A (1) + G (1) + G (1) + A (1) + G (1) = **6 units**.
  - **Result**: PASSES.
- **Measure Pattern 4 (Varies by line):**
  - `| G3 EGc |`: G3(3) + E(1) + G(1) + c(1) = **6 units**. (PASSES)
  - `| G3 EFG |`: G3(3) + E(1) + F(1) + G(1) = **6 units**. (PASSES)
  - `| G3 EAB |`: G3(3) + E(1) + A(1) + B(1) = **6 units**. (PASSES)

**Step 5: Render Final Verdict** The score passed the syntax scan, and every measure correctly sums to the required 6 units. The rhythm is correct.

$\boxed{True}$

**Human Expert**

$\boxed{True}$

Figure 14: Sheet Music Continuation—Case 3. Model: Qwen3-8B-Base. Includes the prompt, model response, Gemini-2.5-Pro verification, and human evaluation result.

## Prompt

You are an ABC notation music composer. Your task is to continue the provided musical piece for four measures.

**Input Music:**

```
M:2/4
L:1/8
K:C
| "C" EE E2 | "G7" DD DE | "C" C2 C2 | "C" E2 (EF) |
```

The new measures **must be a creative variation** of the original melody, not a direct repetition. please enclose the entire ABC notation string within \boxed{}.

## Response

To continue the provided musical piece with a creative variation, we can introduce new melodic ideas while maintaining the rhythm and key. Here's a possible continuation:

```
M:2/4
L:1/8
K:C
| "C" EE E2 | "G7" DD DE | "C" C2 C2 | "C" E2 (EF) |
| "C" FF F2 | "G7" GG GF | "C" D2 D2 | "C" F2 (FG) |
```

**Explanation:**

1. **First Measure:** FF F2 introduces a new note F and maintains the rhythm.
2. **Second Measure:** GG GF introduces a new note G and a descending interval.
3. **Third Measure:** D2 D2 introduces a new note D and maintains the rhythm.
4. **Fourth Measure:** F2 (FG) introduces a new note F and a descending interval.

This continuation provides a creative variation while staying within the same key and rhythm.

Enclosed within \boxed{}:

```
\boxed{
    M:2/4
    L:1/8
    K:C
    | "C" EE E2 | "G7" DD DE | "C" C2 C2 | "C" E2 (EF) |
    | "C" FF F2 | "G7" GG GF | "C" D2 D2 | "C" F2 (FG) |
}
```

## Gemini-2.5-Pro Verification

**Applying the Guideline:**

1. **Step 1: Perform Global Syntax Scan**
    - **Action**: Scan the generated measures: | "C" FF F2 | "G7" GG GF | "C" D2 D2 | "C" F2 (FG) |
    - The score uses only natural notes (C, D, E, F, G). There are no accidentals, and therefore no non-compliant postfix accidentals (like C# or Bb).
    - **Judgment**: The score **PASSES** the syntax check.
2. **Step 2: Establish Rhythmic Framework**
    - **Meter (M:)**: 2/4
    - **Default Note Length (L:)**: 1/8
    - **Required Sum**: (2/4) / (1/8) = 0.5 / 0.125 = 4. Each measure must sum to **4** eighth-note units.
3. **Step 3 & 4: Conduct Measure-by-Measure Analysis**
    - **Measure 1 (FF F2):**
        - F (1) + F (1) + F2 (2) = 4 units.
        - **Judgment**: Correct. 4 == 4.
    - **Measure 2 (GG GF):**
        - G (1) + G (1) + G (1) + F (1) = 4 units.
        - **Judgment**: Correct. 4 == 4.
    - **Measure 3 (D2 D2):**
        - D2 (2) + D2 (2) = 4 units.
        - **Judgment**: Correct. 4 == 4.
    - **Measure 4 (F2 (FG)):**
        - F2 (2) + F (1) + G (1) = 4 units. (Note: Slurs do not affect rhythmic value).
        - **Judgment**: Correct. 4 == 4.
4. **Step 5: Render Final Verdict**
    - The score passes the syntax check, and all generated measures pass the rhythm check.

*True*

## Human Expert

*True*

Figure 15: Sheet Music Continuation—Case 4. Model: Qwen3-8B-Base+GRPO. Includes the prompt, model response, Gemini-2.5-Pro verification, and human evaluation result.

## Prompt

You are an ABC notation music composer. Your task is to continue the provided musical piece for four measures.

**Input Music:**

```
M:4/4
L:1/8
K:A
| efed c4 | e2 f2 =g2 f2 | e4- e4 | e2 f2 =g2 f2 |
```

The new measures **must be a creative variation** of the original melody, not a direct repetition. please enclose the entire ABC notation string within \boxed{}.

## Response

To continue the provided musical piece in ABC notation while maintaining a creative variation of the original melody, I will analyze the rhythm and melody of the given measures and create new ones that follow a similar structure but with variations. Here's the continuation:

```
M:4/4
L:1/8
K:A
| efed c4 | e2 f2 =g2 f2 | e4- e4 | e2 f2 =g2 f2 |
| f2 g2 a2 b2 | c2 d2 e2 f2 | g2 a2 b2 c2 | d2 e2 f2 g2 |
```

**Explanation:**

1. **First Measure:** f2 g2 a2 b2 - This measure introduces higher notes (f, g, a, b) while maintaining the 4/4 time signature.
2. **Second Measure:** c2 d2 e2 f2 - This measure continues the ascending pattern with the notes c, d, e, f.
3. **Third Measure:** g2 a2 b2 c2 - This measure further extends the ascending pattern with the notes g, a, b, c.
4. **Fourth Measure:** d2 e2 f2 g2 - This measure completes the ascending pattern with the notes d, e, f, g.

This continuation maintains the 4/4 time signature and the 1/8 note length, while creatively varying the melody by introducing higher notes and extending the ascending pattern.

Enclosed within \boxed{}:

```
\boxed{
    M:4/4
    L:1/8
    K:A
    | efed c4 | e2 f2 =g2 f2 | e4- e4 | e2 f2 =g2 f2 |
    | f2 g2 a2 b2 | c2 d2 e2 f2 | g2 a2 b2 c2 | d2 e2 f2 g2 |
}
```

## Gemini-2.5-Pro Verification

**Analysis based on the provided guideline:**

**1. Step 1: Perform Global Syntax Scan** * **Action:** The generated measures are | f2 g2 a2 b2 | c2 d2 e2 f2 | g2 a2 b2 c2 | d2 e2 f2 g2 |. A scan of this code reveals no postfix accidentals (like C# or Bb). * **Judgment:** The score **PASSES** the syntax check.

**2. Step 2: Establish Rhythmic Framework** * **Action:** From the header, M:4/4 and L:1/8. * **Calculation:** Required Sum = (4/4) / (1/8) = 1 / (1/8) = 8. Each measure must sum to 8 eighth-note units.

**3. Step 3 & 4: Conduct Measure-by-Measure Analysis** * **Generated Measure 1:** | f2 g2 a2 b2 | * Calculation: f2(2) + g2(2) + a2(2) + b2(2) = 8 units. * Judgment: 8 == 8. The measure **PASSES**. * **Generated Measure 2:** | c2 d2 e2 f2 | * Calculation: c2(2) + d2(2) + e2(2) + f2(2) = 8 units. * Judgment: 8 == 8. The measure **PASSES**. * **Generated Measure 3:** | g2 a2 b2 c2 | * Calculation: g2(2) + a2(2) + b2(2) + c2(2) = 8 units. * Judgment: 8 == 8. The measure **PASSES**. * **Generated Measure 4:** | d2 e2 f2 g2 | * Calculation: d2(2) + e2(2) + f2(2) + g2(2) = 8 units. * Judgment: 8 == 8. The measure **PASSES**.

**5. Step 5: Render Final Verdict** * The score passed the syntax check, and all generated measures passed the rhythmic check.

$\boxed{True}$

## Human Expert

$\boxed{True}$

Figure 16: Sheet Music Continuation—Case 5. Model: Qwen3-8B-Base+GRPO. Includes the prompt, model response, Gemini-2.5-Pro verification, and human evaluation result.

## Prompt

You are an ABC notation music composer. Your task is to continue the provided musical piece for four measures.

**Input Music:**

```
M:4/4
L:1/8
Q:1/8=232
K:D
| d>c d>A F>A d>f | e>d c>B A2 g2 | f>g a>f b>a g>f | e>d c>B A>g f>e |
```

The new measures **must be a creative variation** of the original melody, not a direct repetition. please enclose the entire ABC notation string within \boxed{}.

## Response

To continue the provided musical piece in a creative variation while maintaining the 4/4 time signature and the 1/8 note duration, we can explore different melodic ideas that still feel connected to the original theme. Here's a possible continuation:

```
M:4/4
L:1/8
Q:1/8=232
K:D
| d>c d>A F>A d>f | e>d c>B A2 g2 | f>g a>f b>a g>f | e>d c>B A>g f>e |
| g>f e>d c>B A>G | f>e d>c B>A G>F | E>D C>B A>G F>E | D>C B>A G>F E>D |
```

**Explanation:**

1. **First Measure (g>f e>d c>B A>G):** This measure starts with a descending line from G to F, then continues with a similar pattern from E to D, C to B, and A to G. It maintains the rhythmic flow but introduces a new descending line.

2. **Second Measure (f>e d>c B>A G>F):** This measure starts with a descending line from F to E, then continues with a similar pattern from D to C, B to A, and G to F. It keeps the rhythmic consistency while introducing a new melodic idea.

3. **Third Measure (E>D C>B A>G F>E):** This measure starts with a descending line from E to D, then continues with a similar pattern from C to B, A to G, and F to E. It maintains the rhythmic flow and introduces a new melodic direction.

4. **Fourth Measure (D>C B>A G>F E>D):** This measure starts with a descending line from D to C, then continues with a similar pattern from B to A, G to F, and E to D. It concludes the piece with a smooth descending line, mirroring the original theme but with a creative variation.

This continuation should feel like a natural extension of the original piece while introducing new melodic ideas.

## Gemini-2.5-Pro Verification

**Step 1: Perform Global Syntax Scan** * **Action**: Scan the entire score for invalid postfix accidentals (Bb, C#, etc.). * **Judgment**: No postfix accidentals are found. The score **PASSES** the syntax check.

**Step 2: Establish Rhythmic Framework** * **Action**: Calculate the `Required Sum` of L: units per measure. * `M:4/4` and `L:1/8` * `Required Sum = (4/4) / (1/8) = 1 / (1/8) = 8` units.

**Step 3 & 4: Conduct Measure-by-Measure Analysis** Each broken rhythm group (A>B) is worth `1.5 + 0.5 = 2.0` units. Each note with a multiplier of 2 (A2) is worth `1 * 2 = 2.0` units.

- **Measure 1:** d>c d>A F>A d>f -> $2 + 2 + 2 + 2 = 8$. (Correct)
- **Measure 2:** e>d c>B A2 g2 -> $2 + 2 + 2 + 2 = 8$. (Correct)
- **Measure 3:** f>g a>f b>a g>f -> $2 + 2 + 2 + 2 = 8$. (Correct)
- **Measure 4:** e>d c>B A>g f>e -> $2 + 2 + 2 + 2 = 8$. (Correct)
- **Measure 5:** g>f e>d c>B A>G -> $2 + 2 + 2 + 2 = 8$. (Correct)
- **Measure 6:** f>e d>c B>A G>F -> $2 + 2 + 2 + 2 = 8$. (Correct)
- **Measure 7:** E>D C>B A>G F>E -> $2 + 2 + 2 + 2 = 8$. (Correct)
- **Measure 8:** D>C B>A G>F E>D -> $2 + 2 + 2 + 2 = 8$. (Correct)

**Step 5: Render Final Verdict** The score passes the syntax scan and all measures are rhythmically correct.

$\boxed{True}$

## Human Expert

$\boxed{True}$

Figure 17: Sheet Music Continuation—Case 6. Model: Qwen3-8B-Base+GRPO. Includes the prompt, model response, Gemini-2.5-Pro verification, and human evaluation result.

## Prompt

You are an ABC notation music composer. Your task is to continue the provided musical piece for four measures.

**Input Music:**

```
M:4/4
L:1/8
Q:1/2=60
K:G
| g4 d4 | cBAG GFED | G2 Bd e>deB | A2 A>A A2 B2 |
```

The new measures **must be a creative variation** of the original melody, not a direct repetition. please enclose the entire ABC notation string within \boxed{}.

## Response

```
M:4/4
L:1/8
Q:1/2=60
K:G
| g4 d4 | cBAG GFED | G2 Bd e>deB | A2 A>A A2 B2 | C2 D E F# | G2 A B C |
```

**Explanation of the Creative Variation:**

The original melody consists of four measures, with a clear melodic progression in G major. To create a **creative variation**, I extended the piece with two new measures that maintain the key (G major), tempo (60 BPM), and rhythm (quarter notes and eighth notes), but with **distinct melodic patterns**.

- **New Measure 1:** `C2 D E F#`
  This introduces a rising motif starting with C and ascending through D, E, and F#, which is a new sequence and distinct from the original's G–D and C–B–A–G patterns.

- **New Measure 2:** `G2 A B C`
  This continues the ascending motion with a new sequence of G–A–B–C, forming a diatonic scale in G major, while maintaining the rhythmic structure and avoiding direct repetition of the original's descending or recurring melodic fragments.

These measures are a **creative variation** that respects the original's tonal and rhythmic framework but introduces new musical ideas, ensuring originality and thematic development.

## Gemini-2.5-Pro Verification

**Analysis using the provided guideline:**

1. **Step 1 (Syntax Scan):**
   - The generated measure `| C2 D E F# |` contains the notation `F#`.
   - According to the guideline's "Governing Rule of Pitch Notation", `F#` is an "Invalid Syntax (Non-Compliant Extension)". The compliant syntax is `^F`.
   - **Judgment:** The score is **NON-COMPLIANT** due to syntax. The process could stop here, but for completeness, we will check the rhythm.
2. **Step 2 (Rhythmic Framework):**
   - `M:4/4` and `L:1/8`.
   - Required Sum = (4/4) / (1/8) = 8 units per measure.
3. **Step 3 & 4 (Measure Analysis):**
   - **Generated Measure 1:** `| C2 D E F# |`
     - C2 = 2 units
     - D = 1 unit
     - E = 1 unit
     - F# = 1 unit
     - **Calculated Sum:** $2 + 1 + 1 + 1 = 5$ units.
     - **Judgment:** 5 != 8. The measure **FAILS** the rhythmic check.
   - **Generated Measure 2:** `| G2 A B C |`
     - G2 = 2 units
     - A = 1 unit
     - B = 1 unit
     - C = 1 unit
     - **Calculated Sum:** $2 + 1 + 1 + 1 = 5$ units.
     - **Judgment:** 5 != 8. The measure **FAILS** the rhythmic check.

**Final Verdict:** The generated music fails both the syntax check (due to `F#`) and the rhythmic integrity check for both new measures.

`False`

## Human Expert

`False`

Figure 18: Sheet Music Continuation—Case 7. Model: Qwen3-8B-Thinking. Includes the prompt, model response (excluding content within <think></think>), Gemini-2.5-Pro verification, and human evaluation result.

### Prompt

You are an ABC notation music composer. Your task is to continue the provided musical piece for four measures.

**Input Music:**

```
M:6/8
L:1/8
K:D
| "G" Bcd efg | "D" faf"G" ege | "D" dfd"A" cBA | "D" d3 FGA |
```

The new measures **must be a creative variation** of the original melody, not a direct repetition. please enclose the entire ABC notation string within \boxed{}.

### Response

```
M:6/8
L:1/8
K:D
| "G" Bcd efg | "D" faf"G" ege | "D" dfd"A" cBA | "D" d3 FGA |
| "A" cdefgA | "D" edcE dEg | "D" edfE cBA | "D" e3 FGA |
```

The continuation introduces creative variations by altering the starting note and melodic contours while maintaining the original rhythm and key. The first new measure begins with "A" instead of "G," and the subsequent measures use different note sequences that reflect the original patterns but with distinct melodic elements, ensuring a fresh yet coherent continuation.

### Gemini-2.5-Pro Verification

**Applying the Guideline**

**1. Step 1: Perform Global Syntax Scan** * **Action**: Scan the generated measures: `| "A" cdefgA | "D" edcE dEg | "D" edfE cBA | "D" e3 FGA |` * **Judgment**: No invalid postfix accidentals (like `C#`, `Bb`) are found. The score **PASSES** the syntax check.

**2. Step 2: Establish Rhythmic Framework** * **Action**: Identify `M:6/8` and `L:1/8`. * **Calculation**: Required Sum = (6/8) / (1/8) = **6** units per measure.

**3. Step 3 & 4: Conduct Measure-by-Measure Analysis**

- **Generated Measure 1:** `| "A" cdefgA |`
    - $c(1) + d(1) + e(1) + f(1) + g(1) + A(1) = \mathbf{6}$ units.
    - Calculated Sum (6) == Required Sum (6). This measure **PASSES**.
- **Generated Measure 2:** `| "D" edcE dEg |`
    - $e(1) + d(1) + c(1) + E(1) + d(1) + E(1) + g(1) = \mathbf{7}$ units.
    - Calculated Sum (7) != Required Sum (6). This measure **FAILS**.
- **Generated Measure 3:** `| "D" edfE cBA |`
    - $e(1) + d(1) + f(1) + E(1) + c(1) + B(1) + A(1) = \mathbf{7}$ units.
    - Calculated Sum (7) != Required Sum (6). This measure **FAILS**.
- **Generated Measure 4:** `| "D" e3 FGA |`
    - $e3(3) + F(1) + G(1) + A(1) = \mathbf{6}$ units.
    - Calculated Sum (6) == Required Sum (6). This measure **PASSES**.

**5. Step 5: Render Final Verdict** Two of the four generated measures do not have the correct number of rhythmic units. Therefore, the generated music is rhythmically incorrect.

$\boxed{False}$

### Human Expert

$\boxed{False}$

Figure 19: Sheet Music Continuation—Case 8. Model: Qwen3-8B-Thinking. Includes the prompt, model response (excluding content within <think></think>), Gemini-2.5-Pro verification, and human evaluation result.

### Prompt

You are an ABC notation music composer. Your task is to continue the provided musical piece for four measures.

**Input Music:**

```
M:4/4
L:1/8
Q:1/4=120
K:C
| z2 c2 B2 A2 | ^G8 | A3 B c2 E2 | A2 G2 F4 |
```

The new measures **must be a creative variation** of the original melody, not a direct repetition. please enclose the entire ABC notation string within \boxed{}.

### Response

```
\boxed{
  M:4/4
  L:1/8
  Q:1/4=120
  K:C
  | z2 c2 B2 A2 | ^G8 | A3 B c2 E2 | A2 G2 F4 |
  | F2 E2 D2 C2 | E3 F G2 A2 | ^A8 | C3 D E2 F2 |
}
```

### Gemini-2.5-Pro Verification

**Applying the Guideline:**

1. **Step 1 (Global Syntax Scan):** The generated score uses `^A8` for a sharp. This is the correct prefix notation. No invalid postfix accidentals (like `A#`) are present. The score **PASSES** the syntax check.

2. **Step 2 (Establish Rhythmic Framework):**
   - `M:4/4` (Meter)
   - `L:1/8` (Default Note Length)
   - `Required Sum = (4/4) / (1/8) = 1 / (1/8) = 8` units per measure.

3. **Step 3 & 4 (Measure-by-Measure Analysis of Generated Music):**
   - **Measure 1 (Generated): | F2 E2 D2 C2 |**
     - F2(2) + E2(2) + D2(2) + C2(2) = 8 units.
     - 8 == 8. The measure **PASSES**.
   - **Measure 2 (Generated): | E3 F G2 A2 |**
     - E3(3) + F(1) + G2(2) + A2(2) = 8 units.
     - 8 == 8. The measure **PASSES**.
   - **Measure 3 (Generated): | ^A8 |**
     - ^A8(8) = 8 units.
     - 8 == 8. The measure **PASSES**.
   - **Measure 4 (Generated): | C3 D E2 F2 |**
     - C3(3) + D(1) + E2(2) + F2(2) = 8 units.
     - 8 == 8. The measure **PASSES**.

4. **Step 5 (Final Verdict):** All generated measures are rhythmically correct.

> *True*

### Human Expert

> *True*

Figure 20: Sheet Music Continuation—Case 9. Model: Qwen3-8B-Thinking. Includes the prompt, model response (excluding content within <think></think>), Gemini-2.5-Pro verification, and human evaluation result.

