# OpenReview forum: "Towards an AI Musician: Synthesizing Sheet Music Problems for Musical Reasoning"
_ICLR.cc/2026/Conference — Submitted to ICLR 2026_

### Official Review · Reviewer_D6f4 · 2025-10-27

**Soundness:** 3
**Presentation:** 3
**Contribution:** 2
**Rating:** 6
**Confidence:** 4

**Summary:**

This paper proposes a benchmark for reasoning on synthetic music sheets, leveraging synthetic data created through reinforcement learning. This is combined with experiments across LLMs and MLLMs, also showing how models also have abilities beyond reasoning in music composition.

**Strengths:**

* The problem of reasoning on music sheets is relatively unexplored, and the contribution of leveraging music theory rules to synthesize sheet music is sufficiently novel.
* Experimental results across various LLMs and MLLMs are well constructed and cover textual QA as well as visual QA plus benchmarks on real-world data.

**Weaknesses:**

* Some assumptions made in this work are not clarified in the paper, and some design choices are not presented with sufficient clarity - see the below section for details.
* The resulting synthetic data are still fairly simple from a musical point of view and might not scale to complex compositions, or to noisy and degraded images of scores or even handwritten notation.
* The manuscript includes very few critical remarks arguing on the strengths and weaknesses of the proposed benchmark and offers no clear avenues for future work, therefore limiting the scholarly quality of the paper.

**Questions:**

* Section 1 and 2.1 make several claims that "sheet music is the universal language" which are not correct from a musical or musicological point of view. This work uses a particular form of music notation (namely Western staff notation) but not all music cultures use this type of notation, and not all music cultures use music notation in the first place. Even some styles within Western music use alternative forms of notation, e.g. guitar tabs. Therefore the scope of this work needs to be revised accordingly and certain assumptions made regarding the data (and theory) domain would need to be clarified.
* Some aspects of this work involve converting ABC notation to images. This conversion process can have significant effects on performance for visual QA tasks, however there is no information provided on which notation-to-image engraving method was used, and there appears to not have been any work on using different engraving methods as to improve the diversity of the image data.
* From Figure 4 the composition of the test set appears to cover questions related to Chords, Scale, Intervals, and Rhythm. One question might be: are these 4 categories sufficient to claim understanding and reasoning for sheet music? Which musical aspects might be missing from these questions which could form a new benchmark?
* Table 2: what is the "Thinking" column?
* In section 4.1, there are comparisons with real-world benchmarks. For MMMU Music and the Visual QA task, it appears that the table only shows results using the proposed method. How is performance when compared with reported results and other methods using the MMMU benchmark and specifically its Visual QA task?
* Section 6 only provides a limited summary of the paper. Instead this could have provided some critical remarks on this work, emphasizing on the strengths but also the areas of possible improvement for this work, combined with directions of future work. Currently there is limited critical material on this paper which limits its scholarly contribution.

---

> ### Author Response · Authors · 2025-11-20
> **Author Response (1/4)**
>
> Thank you for your insightful feedback. We will address each of these points individually.
> ### Weakness
>
> 1. Some assumptions made in this work are not clarified in the paper, and some design choices are not presented with sufficient clarity - see the below section for details.
>
>     Thank you for the insightful feedback. We will revise our paper accordingly, as the following responses.
>
> 2. The resulting synthetic data are still fairly simple from a musical point of view and might not scale to complex compositions, or to noisy and degraded images of scores or even handwritten notation.
>
>
>     Thank you for this important comment. Our current framework focuses on the foundational, local rules of music theory. We agree that our initial claim of “full sheet music reasoning” was an overstatement, and we will revise it throughout the manuscript to the more precise term “foundational musical reasoning”.
>
>     This work represents the crucial first step in our research roadmap. We began with these foundational domains because the rules are deterministic, allowing us to develop and validate our data synthesis pipeline with a high degree of confidence.
>
>     For higher-level concepts like harmonic progression or musical form, we can indeed leverage powerful tools like music21 to develop programmatic synthesis algorithms. However, the complexity and stylistic nuance of these rules mean that the correctness of the synthesis algorithms themselves must be rigorously validated.
>
>     Therefore, this next stage requires a thorough black-box testing phase involving a panel of music experts to certify that our synthesis algorithms are musically sound across a wide range of examples. Before committing to this expert-led validation effort, it was imperative for us to first demonstrate that the underlying RLVR training paradigm is effective for musical tasks.
>
>     Regarding the conversion of scores or handwritten notation, we believe this is outside the current scope of our work. Users can independently generate their own style-specific image representations from ABC notation, which can then be used to create diverse datasets. While the exploration of image synthesis and augmentation methods for these images is an interesting and valuable direction, it represents a separate line of research that our work aims to inspire in the future.
>
>     Finally, we present the details of our roadmap, next steps, and technical plan for your review of the feasibility addressed in our response to Weakness 3. We look forward to receiving your insightful feedback.

---

> ### Author Response · Authors · 2025-11-20
> **Author Response (2/4)**
>
> 3. The manuscript includes very few critical remarks arguing on the strengths and weaknesses of the proposed benchmark and offers no clear avenues for future work, therefore limiting the scholarly quality of the paper.
>
>     Thank you for this constructive suggestion. We agree that providing a transparent critique of our current limitations and outlining a clear path forward significantly enhances the scholarly quality of the work.
>     To address this, we will add a new section, "Section 6: Limitations and Future Work," to the revised manuscript. In this section, we critically analyze the current focus on local, rule-based synthetics and share our research roadmap to guide the community:
>
>     ---
>
>     **Foundational Musical Reasoning. (QA) (This work)**
>
>     This initial work was all about ensuring the model masters the fundamental "grammar" of music theory. Before understanding musical ideas, it must correctly identify non-negotiable errors like incorrect rhythm. At the same time, this work aims to explore the feasibility and potential of this direction, offering a new avenue for further research.
>
>     **Higher-Level Structural and Harmonic Reasoning.  (QA) (Next Step)**
>
>     Our next step, which is our current focus, is to teach the model to reason about musical structure. To do this, we will leverage the **Music21** toolkit. Its analytical capabilities allow us to parse sheet music and generate clear, objective "ground truth" data. This enables us to automatically create a series of verifiable questions that rigorously test the model's ability to reason about high-level structures, such as harmonic progressions. At this stage, we also require additional expert professionals to validate the accuracy of the generation algorithm through black-box testing.
>
>     Besides, a key goal of the second stage is to engineer a systematic Python package. This framework will be designed to synthesize musical reasoning problems from low to high levels. We plan to build it with good software engineering principles, create an extensible question types, and invite the research community to contribute. We believe building a true AI musician requires a passionate, collaborative effort.
>
>     **Sheet Music Creation. (Generation & QA) (Final Step)**
>
>     Once the model can reliably analyze and understand musical structure, the final and most exciting stage is to have it start creating its own. This is where we shift from analysis to generation.  In this stage, we will move beyond objective judgment, although we have not yet fully decided on the approach to achieve this.
>
>     **Others**
>
>     While this roadmap outlines our specific path forward, we believe this work can inspire a range of valuable research directions for the community, including:
>
>     1. Extending to develop automated methods for converting standard staff notation into other symbolic formats (e.g., tablature, lead sheets, or ABC notation).
>     2. Designing image synthesis methods that generate scores with human-like stylistic handwriting variations  from abc notation becomes very important and meaningful work. It can help MLLMs to generalize to real-world, "in-the-wild" scenarios, such as analyzing scores from smartphone photos or scanned historical documents.
>     3. Beyond the visual domain, sheet music is a symbolic representation that can be programmatically rendered into other modalities, such as MIDI (performance data) and audio (sound). This allows for the creation of large-scale, perfectly aligned, multi-modal datasets.
>
>     Since the underlying musical logic remains consistent across different notations and modalities, our paradigm could be used to generate parallel corpora of multi-format and multi-modal scores with identical ground-truth data. This verifiable consistency is particularly valuable for RLVR, enabling scalable reinforcement learning of musical understanding across diverse modalities and notations. All these potential research directions can be integrated into our system, making it a foundational tool for the research community.
>
>     ---
>
>     We believe our work can inspire a wide range of future research, and we will include a section that provides detailed guidance for others to follow our approach.

---

> > ### Author Response · Authors · 2025-11-20
> > **Author Response (3/4)**
> >
> > ### Question
> >
> > 1. Section 1 and 2.1 make several claims that "sheet music is the universal language" which are not correct from a musical or musicological point of view. This work uses a particular form of music notation (namely Western staff notation) but not all music cultures use this type of notation, and not all music cultures use music notation in the first place. Even some styles within Western music use alternative forms of notation, e.g. guitar tabs. Therefore the scope of this work needs to be revised accordingly and certain assumptions made regarding the data (and theory) domain would need to be clarified.
> >
> >
> >     Thank you for highlighting this important and precise point. You are absolutely correct from both a musical and musicological perspective. While we believe the broader term "sheet music" can encompass various systems, including the excellent example of guitar tablature you provided, our current framework is specifically designed for the rule-based structure of staff notation. The use of the phrase "universal language" was indeed an oversimplification, and we appreciate you bringing this inaccuracy to our attention. We will revise this accordingly.
> >
> > 2. Some aspects of this work involve converting ABC notation to images. This conversion process can have significant effects on performance for visual QA tasks, however there is no information provided on which notation-to-image engraving method was used, and there appears to not have been any work on using different engraving methods as to improve the diversity of the image data.
> >
> >
> >     To address your first point, we will ensure this is highlighted more clearly in the manuscript. As mentioned on page 3 (line 160), our process is a two-step pipeline:
> >
> >     - First, we use the abcm2ps library to convert the ABC notation into a high-quality vector image.
> >     - Then, we use ImageMagick to render this into a standard image format for the visual QA tasks.
> >
> >      We will make this information more prominent in our experimental setup section to ensure full reproducibility.
> >
> >     On the other hand, you are absolutely correct that for real-world applications, models must generalize to a wide variety of visual renderings. This can be achieved by altering fonts, typefaces, spacing, and so on, as well as incorporating traditional computer vision data augmentation methods to enhance data diversity. However, this was not the focus of our current paper.  We plan to incorporate these aspects into the next tool in our work, as part of our shared research plan.
> >
> > 3. From Figure 4 the composition of the test set appears to cover questions related to Chords, Scale, Intervals, and Rhythm. One question might be: are these 4 categories sufficient to claim understanding and reasoning for sheet music? Which musical aspects might be missing from these questions which could form a new benchmark?
> >
> >
> >     Thank you for your insightful question regarding the scope of question types. Our four categories of Chords, Scale, Intervals, and Rhythm represent the foundational level of musical literacy. While essential, they are not sufficient on their own to constitute comprehensive musical reasoning, which we aim to address in the next phase. In response to weaknesses 2&3, we have outlined our research plan and technical details. We will add a section discussing limitations and future work. We also look forward to collaborating with the research community to advance the development of AI musician.
> >
> > 4. Table 2: what is the "Thinking" column?
> >
> >
> >     The "Thinking" column indicates whether the reasoning mode is activated in a reasoning model. For some models, the reasoning mode can be turned off, and we use the term "thinking" to specify when it is enabled. We acknowledge that this point was not sufficiently clear in the original manuscript and will revise it to better convey this limitation.

---

> ### Author Response · Authors · 2025-11-20
> **Author Response (4/4)**
>
> 5. In section 4.1, there are comparisons with real-world benchmarks. For MMMU Music and the Visual QA task, it appears that the table only shows results using the proposed method. How is performance when compared with reported results and other methods using the MMMU benchmark and specifically its Visual QA task?
>
>
>     | Models | MMMU Music |
>     | --- | --- |
>     | Gemini-2.5-pro | 36.67 |
>     | Qwen2.5-VL-7B-IT | 20.00 |
>     | Qwen2.5-VL-7B-IT+GRPO | 26.67 |
>
>     Thank you for your insightful question. In section 4.1, we present the results of the proposed method on the MMMU Music benchmark. To address your point, we have included the performance of **Gemini-2.5-Pro**, which is one of the most powerful MLLMs currently available. Despite its large scale—potentially hundreds or even thousands of times larger than our 7B model—**Gemini-2.5-Pro** achieves only **36.67%** accuracy on MMMU Music, according to recent benchmarks. This emphasizes that the task is extremely challenging, and even state-of-the-art generalist models struggle to perform well.
>
> 6. Section 6 only provides a limited summary of the paper. Instead this could have provided some critical remarks on this work, emphasizing on the strengths but also the areas of possible improvement for this work, combined with directions of future work. Currently there is limited critical material on this paper which limits its scholarly contribution.
>
>     Thank you for your valuable feedback! We will revise this section to incorporate a more in-depth critique and outline potential future work, as we respond to Weakness 3.

---

### Official Review · Reviewer_yS4X · 2025-10-27

**Soundness:** 3
**Presentation:** 2
**Contribution:** 3
**Rating:** 6
**Confidence:** 4

**Summary:**

This paper introduces a new benchmark that generates sheet-music reasoning problems in text and image form based on core music-theory rules (N=1600 + 8000 each). The main application of this benchmark is to train LLM/VLM to reason about sheet music. The paper shows that using this synthetic data for Reinforcement Learning with Verifiable Rewards improves LLMs performance on the given benchmark, as well as on two existing benchmarks, MusicTheoryBench and MMMU.

**Strengths:**

Programmatic pipeline for synthetic data generation allows a principled generation of a large number of problems, allowing to generate a much larger benchmark compared to existing ones.

The benchmark is novel, and includes diverse question types.

Beyond music, the pattern “encode domain rules -> synthesize verifiable tasks -> RLVR” is general and timely for reasoning-centric training

 The expert-templated, rules-driven generator produces auto-graded, controllable items (bar placement, interval/chord/scale families), and deterministically renders images. This is ideal for RLVR where reward correctness matters.

A wide a range of state-of-the-art LLMs and MLLMs are included in the evaluation

**Weaknesses:**

Presentation clarity.

Dataset only focuses on 4 music theory domains (scale, rhythm, interval, chord). The questions are focused on music theory rule application, wheres for a reasoning benchmark it would be good to consider tasks such as inferring key modulations or harmonic function given a progression, which involve integrating symbolic, structural, and contextual information.

Even a small study comparing how the best-performing fine-tuned models perform compared to humans would be useful.

**Questions:**

Are there existing non-LLM based models that can solve these reasoning problems?

How was the specific given subset of reasoning questions chosen to be included in the benchmark?

---

> ### Author Response · Authors · 2025-11-20
> **Author Response (1/2)**
>
> We have made general revisions to the presentation based on feedback from other reviewers. If there are any further areas that need improvement, please feel free to let us know. We would be more than happy to make additional refinements.
>
> ### Weakness
>
> 1. Dataset only focuses on 4 music theory domains (scale, rhythm, interval, chord). The questions are focused on music theory rule application, wheres for a reasoning benchmark it would be good to consider tasks such as inferring key modulations or harmonic function given a progression, which involve integrating symbolic, structural, and contextual information.
>
>
>     Thank you for this insightful comment. We fully agree that tasks involving key modulation and harmonic function are essential for fostering comprehensive musical reasoning.
>
>     However, we prioritized the four foundational domains (scale, rhythm, interval, chord) in this initial work because their rules are deterministic. This allowed us to develop and validate our data synthesis pipeline with a high degree of confidence without ambiguity.
>
>     For higher-level concepts like harmonic progression or musical form, we can indeed leverage powerful tools like music21 to develop programmatic synthesis algorithms. However, the complexity and stylistic nuance of these rules mean that the *correctness of the synthesis algorithms themselves must be rigorously validated.*
>
>     Before committing to this resource-intensive expert validation, it was imperative for us to first demonstrate that the underlying RLVR training paradigm is effective for simpler musical tasks. Having successfully demonstrated this feasibility in the current work, we are now actively organizing the expert-led validation required for the next stage of high-level reasoning.
>
>     To address the reviewers' concern, we share our research roadmap. This paper represents the necessary first stage, and we are already actively working on the second stage:
>
>     ---
>
>     **Foundational Musical Reasoning. (QA) (This work)**
>
>     This initial work was all about ensuring the model masters the fundamental "grammar" of music theory. Before understanding musical ideas, it must correctly identify non-negotiable errors like incorrect rhythm. At the same time, this work aims to explore the feasibility and potential of this direction, offering a new avenue for further research.
>
>     **Higher-Level Structural and Harmonic Reasoning.  (QA) (Next Step)**
>
>     Our next step, which is our current focus, is to teach the model to reason about musical structure. To do this, we will leverage the **Music21** toolkit. Its analytical capabilities allow us to parse sheet music and generate clear, objective "ground truth" data. This enables us to automatically create a series of verifiable questions that rigorously test the model's ability to reason about high-level structures, such as harmonic progressions. At this stage, we also require additional expert professionals to validate the accuracy of the generation algorithm through black-box testing.
>
>     Besides, a key goal of the second stage is to engineer a systematic Python package. This framework will be designed to synthesize musical reasoning problems from low to high levels. We plan to build it with good software engineering principles, create an extensible question types, and invite the research community to contribute. We believe building a true AI musician requires a passionate, collaborative effort.
>
>     **Sheet Music Creation. (Generation & QA) (Final Step)**
>
>     Once the model can reliably analyze and understand musical structure, the final and most exciting stage is to have it start creating its own. This is where we shift from analysis to generation.  In this stage, we will move beyond objective judgment, although we have not yet fully decided on the approach to achieve this.
>
>     ---
>
>     We believe our work has the potential to inspire a broad spectrum of future research. To highlight this, we will add a "Limitations and Future Work" section to provide a clear roadmap for upcoming directions.

---

> > ### Author Response · Authors · 2025-11-20
> > **Author Response (2/2)**
> >
> > 2. Even a small study comparing how the best-performing fine-tuned models perform compared to humans would be useful.
> >
> >
> >     Thank you for the valuable suggestion. We carefully considered including a human control study. However, it is important to note that the tasks in our dataset (e.g., interval identification, rhythm counting) correspond to axiomatic music theory rules where the "ground truth" is mathematically defined.
> >
> >     For a trained musician, these tasks are foundational, and human performance effectively hits a "ceiling" of near 100% accuracy. Consequently, a quantitative comparison would not yield significant statistical insight. The core challenge addressed in this paper is not whether the model can purely match human accuracy, but whether it can internalize the reasoning process of a musician.
> >
> >     To illustrate this, we present a case from our Qwen2.5-7B-Base+GRPO model. As shown below, the model does not simply guess the answer; it simulates the exact cognitive workflow a human musician uses to analyze a score:
> >
> >     **example**:
> >
> >     ```
> >     L:1/8
> >     Q:1/4=120
> >     M:3/4
> >     K:F
> >     f4 F2 g2 gg gg g4 G2 a2 ba gf
> >     ```
> >
> >     **model response**:
> >
> >     ```
> >     To determine the correct placement of bar lines in the given sequence of notes, we need to consider the time signature, which is 3/4 in this case. This means each measure must contain exactly three quarter notes (or the equivalent).
> >
> >     Let's analyze the sequence of notes:
> >
> >     L:1/8
> >     Q:1/4=120
> >     M:3/4
> >     K:F
> >     f4 F2 g2 gg gg g4 G2 a2 ba gf
> >
> >     Breaking it down into smaller segments:
> >
> >     1. f4 F2
> >     2. g2 gg gg
> >     3. g4 G2
> >     4. a2 ba gf
> >
> >     Each segment should contain exactly three quarter notes. Let's count the quarter notes in each segment:
> >
> >     1. f4 F2: 4 (f) + 2 (F) = 6 eighth notes
> >     2. g2 gg gg: 2 (g) + 2 (gg) + 2 (gg) = 6 eighth notes
> >     3. g4 G2: 4 (g) + 2 (G) = 6 eighth notes
> >     4. a2 ba gf: 2 (a) + 2 (ba) + 2 (gf) = 6 eighth notes
> >
> >     Since the time signature is 3/4, each measure must contain exactly three quarter notes. Therefore, the correct placement of bar lines would be after the first and third segments, as they each contain exactly three quarter notes.
> >
> >     The correct option is:
> >
> >     A. L:1/8
> >     Q:1/4=120
> >     M:3/4
> >     K:F
> >     | f4 F2 | g2 gg gg | g4 G2 | a2 ba gf |
> >
> >     So, the answer is \boxed{A}.
> >     ```
> >
> >     These reasoning abilities are foundational skills, essential for LLMs to compose sheet music in a manner akin to human musicians.
> >
> >
> > ### Question
> >
> > 1. Are there existing non-LLM based models that can solve these reasoning problems?
> >
> >     Thank you for this question. Yes, traditional symbolic algorithms and encoder-only models (like BERT) can indeed solve these specific multiple-choice questions with high accuracy.
> >
> >     However, the objective of this work is not merely to calculate the correct answers, but to enable LLMs to "think" like musicians. While discriminative models are efficient, they cannot explain why an answer is correct in natural language, nor can they apply that knowledge to generate new music.
> >
> >     By training an LLM to internalize these rules through explicit reasoning rather than simple classification, we aim to bridge the gap between rigid rule-based systems and the creative, generative capabilities required for future AI musicians.
> >
> >
> > 2. How was the specific given subset of reasoning questions chosen to be included in the benchmark?
> >
> >     Thank you for your question. We randomly sampled 400 questions from each category, resulting in a test set of 1,600 questions. The remaining questions were used to construct the training set. Both datasets can be converted into a visual QA format.

---

> > > ### Comment · Reviewer_yS4X · 2025-11-24
> > > **Thank you for the response.**
> > >
> > > I retain my score recommending acceptance.

---

### Official Review · Reviewer_CAkk · 2025-10-31

**Soundness:** 2
**Presentation:** 2
**Contribution:** 3
**Rating:** 4
**Confidence:** 4

**Summary:**

This paper proposes a synthetic data generation framework that converts core music theory rules (rhythm, interval, chord, scale) into programmable functions for producing verifiable sheet-music reasoning problems. The authors introduce SSMR-Bench, consisting of 3,200 evaluation QA pairs (text + image) and a larger 16K training set for reinforcement learning with verifiable rewards (RLVR). They show that training on synthetic musical reasoning data improves performance on SSMR-Bench, and also transfers to existing human-crafted benchmarks such as MusicTheoryBench and MMMU-Music. The paper further claims that the learned reasoning ability benefits symbolic music composition.

**Strengths:**

Novel data synthesis idea.
Treating music theory as a rule-based generator of reasoning problems is conceptually parallel to math problem synthesis, and enables scalable, verifiable QA generation.

Benchmark + training pipeline.
The paper contributes both an evaluation benchmark and a synthetic training dataset, filling a gap for sheet-music reasoning tasks.

Demonstrated improvements on multiple models.
RLVR-trained Qwen models show clear gains on SSMR-Bench and moderate gains on MusicTheoryBench and MMMU-Music.

Cross-domain effect is interesting.
The observation that music reasoning training may improve math reasoning hints at shared symbolic reasoning structure.

**Weaknesses:**

Music theory coverage is limited.
The synthesis only covers local rules (rhythm, interval, chord, scale) and does not address global musical structure, harmonic progression, voice-leading, or form—so the claim of “full sheet music reasoning” is overstated.

Performance still far from SOTA baselines.
Although training improves scores, the models still lag well behind state-of-the-art on MusicTheoryBench, suggesting limited real-world transfer.

MMMU results close to random guessing.
On MMMU-Music, accuracy remains near the 25% random baseline, indicating the model may mainly learn QA format rather than visual score reasoning.

Composition evaluation is narrow.
The “improved music composition” result only tests 4-bar rhythmic alignment with prompt tempo, ignoring key musical elements such as harmonic coherence, chord progression, or longer-term structure.

Minor but NOTABLE typo.
“GRPO” is repeatedly misspelled as GPRO in Lines 208–216.

**Questions:**

Music theory scope
Do you plan to extend synthesis to higher-level rules (e.g., cadences, phrase structure, functional harmony, chord progression)? If not, how should readers interpret the term sheet music reasoning?

Math transfer claim
In Sec. 4.2, you suggest music reasoning helps math reasoning. Can you test whether music+math mixed training outperforms “math-only extra epochs”?

Benchmark gap
Why do models still underperform SOTA on MusicTheoryBench despite large synthetic training? Is the gap due to missing global theory features?

MMMU near-random accuracy
Can you provide an ablation to show whether the model is learning symbolic reasoning or just answer-pattern priors?

Composition evaluation
Would you consider adding harmonic correctness or chord adherence metrics, instead of only rhythm-based alignment?

Annotation correctness
Are the synthetic questions automatically validated? If so, what is the estimated error rate of the generation pipeline?

---

> ### Author Response · Authors · 2025-11-20
> **Author Response (1/4)**
>
> Thank you for your insightful feedback. We will address each of these points individually.
>
> ### Weakness
>
> 1. Music theory coverage is limited. The synthesis only covers local rules (rhythm, interval, chord, scale) and does not address global musical structure, harmonic progression, voice-leading, or form—so the claim of “full sheet music reasoning” is overstated.
>
>
>     Thank you for this important comment. You are correct that our current framework focuses on the foundational, local rules of music theory. We agree that our initial claim of “full sheet music reasoning” was an overstatement, and we will revise it throughout the manuscript to the more precise term **“foundational musical reasoning”.**
>
>     This work represents the crucial first step in our research roadmap. We began with these foundational domains because the rules are deterministic, allowing us to develop and validate our data synthesis pipeline with a high degree of confidence.
>
>     For higher-level concepts like harmonic progression or musical form, we can indeed leverage powerful tools like music21 to develop programmatic synthesis algorithms. However, the complexity and stylistic nuance of these rules mean that the **correctness of the synthesis algorithms themselves must be rigorously validated.**
>
>     Therefore, this next stage requires a thorough **black-box testing phase involving a panel of music experts** to certify that our synthesis algorithms are musically sound across a wide range of examples. Before committing to this expert-led validation effort, it was imperative for us to first demonstrate that the underlying RLVR training paradigm is effective for musical tasks.
>
>     In this work, we believe we have successfully demonstrated this, and we present the detailed roadmap, next steps, and technical specifics in response to Question 1.
>
> 2. Performance still far from SOTA baselines. Although training improves scores, the models still lag well behind state-of-the-art on MusicTheoryBench, suggesting limited real-world transfer.
>
>
>     We thank the reviewer for the insightful question. To put our results in context, GPT-4 is a massive, proprietary model with unmatched generalist reasoning capabilities. In contrast, our work demonstrates the efficiency of reinforcement learning on a smaller 8B open-source model using a modest amount of synthetic data, achieving performance competitive with GPT-4. More importantly, our fine-tuned model significantly outperforms its own instruction-tuned counterpart (e.g., Qwen3-8B-Thinking), showing that our approach effectively enhances its domain-specific skills.

---

> ### Author Response · Authors · 2025-11-20
> **Author Response (2/4)**
>
> 3. MMMU results close to random guessing. On MMMU-Music, accuracy remains near the 25% random baseline, indicating the model may mainly learn QA format rather than visual score reasoning.
>
>
>     |  | MMMU Music |
>     | --- | --- |
>     | Gemini-2.5-pro | 36.67 |
>     | Qwen2.5-VL-7B-IT | 20.00 |
>     | Qwen2.5-VL-7B-IT+GRPO | 26.67 |
>
>     Thank you for this sharp observation regarding the MMMU results. You are correct that the performance, even after training, is modest in absolute terms. However, we argue that this is not due to the model merely learning the QA format, but is instead a realistic measure of the profound difficulty of this specific benchmark, which validates the need for our work.
>
>     To properly contextualize these results, it is crucial to consider the performance of state-of-the-art MLLMs. As recent benchmarks show, **Gemini-2.5-Pro**, a model might be hundreds or even thousands of times larger than our 7B model, achieves only **36.67%** accuracy on MMMU-Music. This highlights that this is an exceptionally challenging task that even the most powerful generalist models struggle with.
>
>     This industry-wide difficulty is, we argue, a direct symptom of the very problem our paper addresses: **the scarcity of large-scale, verifiable training data for sheet music reasoning.** Regarding the concern that the model is only learning the format, our **6.67% absolute improvement** (from 20.00% to 26.67%) demonstrates a genuine, albeit nascent, enhancement in reasoning.
>
>     To further illustrate the symbolic reasoning, consider the following case from Qwen2.5-VL-7B-IT+GRPO. While we are unable to include the image, we will present the question and the response:
>
>     > Q: <image 1> The second measure of the above example uses which rhythmic device?
>     >
>     >
>     >        A. 'Anacrusis' B. 'Hemiola'  C. 'Suspension'  D. 'Augmentation'
>     >
>
>     > A: To determine the correct answer, let's analyze the rhythm in the second measure of the given musical example:
>     >
>     >
>     > The first two notes in the second measure are on the third beat of the measure, and the next two notes are on the fourth beat of the measure. This creates a pattern where the first two beats of the measure are divided into three notes instead of the usual two.
>     >
>     > This is characteristic of a hemiola, where a regular meter is temporarily altered by having a group of three notes where there would normally be only two.
>     >
>     > Therefore, the correct answer is:
>     > \boxed{B}
>     >
>
>     Ultimately, these findings—both the improvement and the remaining challenge—validate our methodology and illuminate a clear direction for future research.
>
> 4. Composition evaluation is narrow. The “improved music composition” result only tests 4-bar rhythmic alignment with prompt tempo, ignoring key musical elements such as harmonic coherence, chord progression, or longer-term structure.
>
>
>     Thank you for this excellent suggestion. To address your point, we have added a new evaluation for harmonic correctness using the **Chord Tone to non-Chord Tone Ratio (CTnCTR)[1,2]**. This metric calculates the percentage of a melody that consists of either stable chord tones or smoothly resolving non-chord tones.
>
>      $$\text{CTnCTR} = \frac{n_c + n_p}{\text{Total Notes}}$$
>
>     - $n_c$ (Chord Tones): Notes belonging to the current triad (Root, 3rd, 5th).
>     - $n_p$ (Proper Non-Chord Tones): Non-chord notes that are exactly 2 semitones (a whole step) away from the immediate next note.
>
>     We present the results, along with our findings on rhythm consistency, below.
>
>     |  | CTnCTR | RC |
>     | --- | --- | --- |
>     | Qwen3-8B-Base | 88.03% | 59.00% |
>     | Qwen3-8B-Base+GRPO Music | **91.21%** | **76.00%** |
>     | Qwen3-8B-Thinking | 88.11% | 61.50% |
>
>     The results show that our model trained with synthetic music data (+GRPO Music) achieves the highest score, indicating that our training approach also enhances the model's ability to generate harmonically coherent melodies.
>
>     [1]Wu S, Yang Y, Wang Z, et al. Generating chord progression from melody with flexible harmonic rhythm and controllable harmonic density[J]. EURASIP Journal on Audio, Speech, and Music Processing, 2024, 2024(1): 4.
>
>     [2]Wu S, Wang Y, Li X, et al. Melodyt5: A unified score-to-score transformer for symbolic music processing[J]. arXiv preprint arXiv:2407.02277, 2024.
>
> 5. Minor but NOTABLE typo. “GRPO” is repeatedly misspelled as GPRO in Lines 208–216
>
>     Thank you for pointing this out. We have corrected the typo in the revised manuscript.

---

> ### Author Response · Authors · 2025-11-20
> **Author Response (3/4)**
>
> ### Question
>
> 1. Music theory scope Do you plan to extend synthesis to higher-level rules (e.g., cadences, phrase structure, functional harmony, chord progression)? If not, how should readers interpret the term sheet music reasoning?
>
>     Yes, we do have a plan. In fact, we appreciate your and the other reviewers' interest in our future plans and are happy to share our research roadmap.
>
>     ---
>     **Foundational Musical Reasoning. (QA) (This work)**
>
>     This initial work was all about ensuring the model masters the fundamental "grammar" of music theory. Before understanding musical ideas, it must correctly identify non-negotiable errors like incorrect rhythm. We see this as the essential groundwork, creating a solid, verifiably correct foundation upon which all future capabilities will be built.
>
>     **Higher-Level Structural and Harmonic Reasoning.  (QA) (Next Step)**
>
>     Our next step, which is our current focus, is to teach the model to reason about musical structure. To do this, we will leverage the **Music21** toolkit. Its analytical capabilities allow us to parse sheet music and generate clear, objective "ground truth" data. This enables us to automatically create a series of verifiable questions that rigorously test the model's ability to reason about high-level structures, such as harmonic progressions. At this stage, we also require additional expert professionals to validate the accuracy of the generation algorithm through black-box testing.
>
>     Besides, a key goal of the second stage is to engineer a systematic Python package. This framework will be designed to synthesize musical reasoning problems from low to high levels. We plan to build it with good software engineering principles, create an extensible question types, and invite the research community to contribute. We believe building a true AI musician requires a passionate, collaborative effort.
>
>     **Sheet Music Creation. (Generation & QA) (Final Step)**
>
>     Once the model can reliably analyze and understand musical structure, the final and most exciting stage is to have it start creating its own. This is where we shift from analysis to generation.  In this stage, we will move beyond objective judgment, although we have not yet fully decided on the approach to achieve this.
>
>     **Others**
>
>     While this roadmap outlines our specific path forward, we believe this work can inspire a range of valuable research directions for the community, including:
>
>     1. Extending to develop automated methods for converting standard staff notation into other symbolic formats (e.g., tablature, lead sheets, or ABC notation).
>     2. Designing image synthesis methods that generate scores with human-like stylistic handwriting variations from abc notation becomes very important and meaningful work. It can help MLLMs to generalize to real-world, "in-the-wild" scenarios, such as analyzing scores from smartphone photos or scanned historical documents.
>     3. Beyond the visual domain, sheet music is a symbolic representation that can be programmatically rendered into other modalities, such as MIDI (performance data) and audio (sound). This allows for the creation of large-scale, perfectly aligned, multi-modal datasets.
>
>     Since the underlying musical logic remains consistent across different notations and modalities, our paradigm could be used to generate parallel corpora of multi-format and multi-modal scores with identical ground-truth data. This verifiable consistency is particularly valuable for RLVR, enabling scalable reinforcement learning of musical understanding across diverse modalities and notations. All these potential research directions can be integrated into our system, making it a foundational tool for the research community.
>
> ---
>
>    We believe our work can inspire a wide range of future research, and we will include a section that provides detailed guidance for others to follow our approach.

---

> > ### Author Response · Authors · 2025-11-20
> > **Author Response (4/4)**
> >
> > 2. Math transfer claim In Sec. 4.2, you suggest music reasoning helps math reasoning. Can you test whether music+math mixed training outperforms “math-only extra epochs”?
> >
> >     Thank you for your excellent question.  Following your suggestion, we have conducted a new experiment in which we trained the Qwen3-8B-Base on a mix of our synthetic music data and math data (+GRPO (Math+Music)).
> >
> >     | Models | Math Benchmarks | MusicTheoryBench |
> >     | --- | --- | --- |
> >     | Qwen3-8B-Base | 33.90 | 28.65 |
> >     |  +GRPO (Music Only) | 43.97 | 49.97 |
> >     |  +GRPO (Math Only) | 52.84 | 33.81 |
> >     |  +GRPO (Math+Music) | 52.13 | 37.38 |
> >
> >     These results show:
> >
> >     1. **Math Benchmarks**: Mixed training (52.13%) does not outperform math-only training (52.84%), suggesting that more high-quality math data is most effective for improving mathematical reasoning.
> >     2. **MusicTheoryBench**: The mixed model (37.38%) outperforms math-only training (33.81%), showing the value of including music data for music tasks.
> >
> >     While music reasoning improves a subset of these mathematical skills (e.g., for rhythm and intervals), training on math data already provides a strong base for these. Therefore, incorporating music data is more likely to enhance domain-specific expertise rather than further improve mathematical performance. We will revise this section and provide a more detailed analysis.
> >
> > 3. Benchmark gap Why do models still underperform SOTA on MusicTheoryBench despite large synthetic training? Is the gap due to missing global theory features?
> >
> >
> >     We have addressed this concern in detail in our response to Weakness 2. In summary, the performance gap is primarily attributable to the vast difference in model scale (our 8B parameter model vs. the proprietary GPT-4). However, our results demonstrate that our method yields significant improvements over the base model, verifying the efficacy of our training paradigm even with a smaller parameter budget.
> >
> >
> > 4. MMMU near-random accuracy Can you provide an ablation to show whether the model is learning symbolic reasoning or just answer-pattern priors?
> >
> >
> >     Please refer to our detailed analysis and case study in the response to Weakness 3. We have provided a comparative analysis with state-of-the-art models (Gemini-2.5-Pro) and a specific qualitative example (Hemiola analysis) to demonstrate that the model is indeed performing symbolic reasoning, and that the low absolute scores reflect the extreme difficulty of the benchmark itself.
> >
> > 5. Composition evaluation Would you consider adding harmonic correctness or chord adherence metrics, instead of only rhythm-based alignment?
> >
> >     We have implemented this excellent suggestion and added a new evaluation metric, the Chord Tone to non-Chord Tone Ratio (CTnCTR), to assess harmonic correctness. Please refer to the table and detailed results presented in our response to Weakness 4, which confirm that our model achieves superior harmonic coherence compared to the baselines.
> >
> > 6. Annotation correctness Are the synthetic questions automatically validated? If so, what is the estimated error rate of the generation pipeline?
> >
> >     The primary validation of our data comes from its construction method. Since the questions are synthesized using deterministic code (based on strict music theory rules), the ground truth is "correct by design," assuming the underlying logic is bug-free.
> >
> >     To verify this implementation, we performed a manual audit of 25 randomly sampled questions per category, which yielded 100% accuracy (0 errors). This manual check serves as a sanity check, confirming that our programmatic generation pipeline is functioning as intended.
> >
> >
> > We sincerely appreciate your time and effort in reviewing our submission and our response, and we are happy to address any additional questions or concerns you may have.

---

> > > ### Comment · Reviewer_CAkk · 2025-11-20
> > >
> > > Thanks for the timely reply and the detail explanation on the methodology and following experiments. However, I still subjectively think the current quality without more following discovery is marginal to ICLR main conference, and I would like to keep my score unchanged.

---

> ### Author Response · Authors · 2025-11-21
> **Author Response**
>
> We sincerely appreciate your continued engagement and your acknowledgement of our detailed response and additional experiments. We have updated the PDF to reflect your suggestions.
>
> While we respect your perspective regarding the depth of our contributions, we would like to offer a final clarification on why we believe this work provides a substantial and necessary value to the ICLR community:
>
> **Our work establishes a new paradigm to address data scarcity in sheet music reasoning.**
>
> Current LLMs in sheet music reasoning is bottlenecked by a severe lack of high-quality, reasoning-intensive training data, which is prohibitively expensive to annotate by experts. While recent advances in RLVR illustrate the potential to incentivize an LLM's inherent reasoning ability, the field lacks the verifiable musical questions necessary to apply this technique in practice.
>
> To bridge this gap, we propose a deployable data synthesis paradigm. By programmatically constructing questions from music theory rules, we transform abstract musical logic into scalable, verifiable computational problems derived from existing sheet music. This represents a **"zero-to-one"** contribution, providing a tangible and cost-effective solution to enhance LLM reasoning in sheet music.
>
> We believe this paradigm serves as a **foundational enabler** for the community. It transforms "music reasoning" from a theoretical challenge into a solvable engineering task, making it an essential technology for the advancement of AI Musicianship.
>
> Thank you again for your time and valuable feedback. We look forward to solving more questions for you.

---

> > ### Comment · Reviewer_CAkk · 2025-11-24
> >
> > Thanks the author. Despite the authors’ thorough and timely responses, I am keeping my score at 4 because I believe the current work remains marginal for acceptance at ICLR in its present form.
> >
> > The proposed paradigm for synthesizing foundational music-theory reasoning data is interesting and clearly fills an existing gap, and the authors’ additional analyses (harmonic evaluation, MMMU explanation, extended roadmap, and mixed-training experiment) strengthened the submission. However, the core limitations I highlighted—restricted music-theory coverage lead to limited applications to musicology and music industry, limited real-world transfer to established benchmarks, and modest improvements on challenging multimodal tasks compared to state-of-te-art—remain only partially addressed and would require further discovery and validation to reach the level of maturity expected for the ICLR main conference.
> >
> > While I appreciate the potential of the proposed direction, the current contribution is still at an early stage. And my overall assessment of “marginally below acceptance” remains unchanged as my subjective opinion, and leave the decision to be made by meta-reviewer.

---

> > > ### Author Response · Authors · 2025-12-02
> > > **Author Response**
> > >
> > > Thank you for recognizing the potential of our proposed direction! We plan to continue building on this philosophy, expanding the framework to cover a broader range of music theories and modalities in our next work.
> > >
> > > We look forward to your attention and feedback once our new work is published.

---

### Official Review · Reviewer_u888 · 2025-11-10

**Soundness:** 2
**Presentation:** 2
**Contribution:** 2
**Rating:** 2
**Confidence:** 3

**Summary:**

The paper proposes a programmatic framework for generating verifiable sheet-music QA paris in text and image modalities. The authors benchmark multiple LLMs and VLMs (zero-shot and fine-tuned with GRPO) on 4 types of questions and music composition.

**Strengths:**

- The paper provides a clean pipeline to create paired text/image questions, which makes evaluation more controllable and reproducible than LLM-written questions.
- The benchmark is balanced across several categories (rhythm, scales, intervals, basic chords) and provide evaluations on text and visual modalities.
- The author further explore if fine-tune with their music sheet qa dataset can improve music composition.

**Weaknesses:**

- The main contribution is the data synthesis framework. But the authors merely scratch the surface in section 2.2 and lots of core details are missing. First of all, does the framework generate music or just the questions for real music sheets with ground truth? Base on my reading on the paper and figure 2, it only generates questions for annotated music data with ground truth (ABC notation). While the authors highlight their framework is built on music theory, there is only one example and it is not clear what music theory are used. In fact, it is simply removing the answers (e.g., bar) in the music sheets based on what the question is asking. I don't think it is appropriate to claim this framework or synthetic data to be “boundless” or “endless”.
- Although the paper claims to address musical reasoning, the current questions do not appear to require genuine reasoning. According to Appendix A, all questions are recognition/identification of local features (e.g., scale, chord, interval). That fits “sheet-music understanding,” but it is not reasoning in the stronger sense as in Yue et al. 2024.
- The empirical results are unsurprising and offer limited insight. (1) Reading ABC is easier than OMR, expected. (2) A reasoning-tuned model (e.g., DeepSeek-R1) outperforming a large general LLM (e.g., DeepSeek-V3) is also expected. (3) Fine-tuning on music-sheet QA improves in-distribution performance, again expected. More importantly, despite introducing four task categories, the paper provides little analysis about which models work better for which tasks (and why), nor does it disentangle recognition failures from theory/logic errors.

**Questions:**

1. What is thinking in table 2 and table 3? Do you mean reasoning models? Explain what it means in the paper.
2. In Section 4.2 the numbers in the main text is different with the numbers in figure 5, which one is correct?
3. What is the meaning of Section 4.2, showing that the music sheet data can improve reasoning in Math? What is the intuition behind and why is it relevant to the paper?

---

> ### Author Response · Authors · 2025-11-20
> **Author Response (1/3)**
>
> Thank you for your thoughtful feedback and careful review of our paper. We appreciate the opportunity to clarify the contributions and details of our data synthesis framework.
>
> ### Weakness
>
> 1. The main contribution is the data synthesis framework. But the authors merely scratch the surface in section 2.2 and lots of core details are missing. First of all, does the framework generate music or just the questions for real music sheets with ground truth? Base on my reading on the paper and figure 2, it only generates questions for annotated music data with ground truth (ABC notation). While the authors highlight their framework is built on music theory, there is only one example and it is not clear what music theory are used. In fact, it is simply removing the answers (e.g., bar) in the music sheets based on what the question is asking. I don't think it is appropriate to claim this framework or synthetic data to be “boundless” or “endless”.
>
>     Thank you for your detailed and insightful feedback. You have correctly identified several areas where our description of the data synthesis framework needs greater clarity and detail. We will address these important points thoroughly in our revision.
>
>     **1. On the Framework's Core Function (Generating Questions, not Music)**
>
>     You are correct in your assessment that our framework **generates verifiable questions from existing, real-world sheet music**, rather than synthesizing the music itself. This is a deliberate design choice central to our primary goal: creating a resource for the **Reinforcement Learning with Verifiable Rewards (RLVR)** paradigm. Our objective is to leverage the vast corpus of human-composed music to create a scalable supply of structured reasoning problems, which by definition requires a ground-truth source.
>
>     **2. On the Use of Music Theory and Lack of Detail**
>
>     We acknowledge that the single "bar placement" example was insufficient to convey the breadth of music theory integrated into our framework. The framework is built upon programmatic functions that encode rules from four fundamental domains: **Rhythm, Chord, Interval, and Scale**, which encompass nine distinct question templates. For instance, our "NoteCompletionByInterval" template programmatically applies the rules of interval calculation, while "ChordKeyRootIdentification" tests the principles of triadic harmony.
>
>     To provide the necessary detail without exceeding the page limits of the main text, we will add a new, comprehensive section to Appendix A.1. This section will fully describe the algorithm and the specific music theory rules encoded in each of the nine question templates. We will then explicitly reference this appendix in Section 2.2 to ensure the core mechanics of our framework are clear and transparent.
>
>     **3. On the Claim of "Boundless" or "Endless"**
>
>     We appreciate you pointing out this overstatement. Our intention was to draw an analogy to mathematics, where a finite set of simple operations (e.g., addition) can generate a virtually infinite number of unique problems from the universe of numbers. Similarly, our framework applies a core set of music theory rules to a massive corpus of sheet music (e.g., the MelodyHub dataset).
>
>     However, you are right that the pool of existing music is finite, making the term "boundless" imprecise. We will revise our claim throughout the paper to more accurately describe the framework's output as **"highly scalable".**

---

> ### Author Response · Authors · 2025-11-20
> **Author Response (2/3)**
>
> 2. Although the paper claims to address musical reasoning, the current questions do not appear to require genuine reasoning. According to Appendix A, all questions are recognition/identification of local features (e.g., scale, chord, interval). That fits “sheet-music understanding,” but it is not reasoning in the stronger sense as in Yue et al. 2024.
>
>     Thank you for this insightful point. It seems the paper link was inadvertently omitted. We assume you are referring to the MMMU benchmark paper.
>
>     You have articulated a key philosophical distinction about the nature of reasoning versus understanding. The most intuitive way to distinguish these tasks from simple recognition is to  to see them as **applied math problems presented in a musical context.**  A model cannot arrive at the correct answer through passive pattern matching; it must execute a series of computational steps.
>
>     In fact, unlike in domains like pure mathematics or logic, where people can often separate a conceptual understanding problem from a procedural reasoning problem, we argue that in sheet music, understanding and reasoning are deeply intertwined. At the foundational level, the act of understanding is the successful execution of this rule-based reasoning.
>
>     Regarding the bar placement example, while the task's description may seem simple (removing bar lines), the reasoning process required to solve it is fundamental to musical comprehension. For a model—or a human student—to correctly place the bar lines in a piece like the one below:
>
>     ```
>     L:1/8
>     M:4/4
>     K:D
>     ed BG A/B/A/G/ ED GB AG d2 d d/d/ ed BG A/B/A/G/ ED GB A/G/A G2 G d/d/
>     ```
>
>     It must perform a series of deductive steps:
>
>     1. **Identify the Time Signature (M:):** The music is in 4/4 time, meaning each measure must contain the equivalent of four quarter notes.
>     2. **Identify the Default Note Length (L:):** The default unit is an eighth note (1/8).
>     3. **Calculate Measure Duration:** The model must calculate that a 4/4 measure requires eight 1/8th note units ((4/4) / (1/8) = 8).
>     4. **Sum Note Durations:** It must then parse the sequence of notes, correctly calculating the duration of each (e.g., ed = 2 units, A/B/A/G/ = 2 units, d2 = 2 units) and sum them. A bar line is placed only when the sum for a measure reaches exactly 8.
>
>     This task demands a model to learn and apply core rules of rhythm and meter. It is not a simple pattern-matching task but a procedural reasoning problem, analogous to skills tested in standardized music theory curricula (e.g., ABRSM).
>
>     We present an example of model reasoning to make this process clearer.
>
>     **example**:
>
>     ```
>     L:1/8
>     Q:1/4=120
>     M:3/4
>     K:F
>     f4 F2 g2 gg gg g4 G2 a2 ba gf
>     ```
>
>     **model response**:
>
>     ```
>     To determine the correct placement of bar lines in the given sequence of notes, we need to consider the time signature, which is 3/4 in this case. This means each measure must contain exactly three quarter notes (or the equivalent).
>
>     Let's analyze the sequence of notes:
>
>     L:1/8
>     Q:1/4=120
>     M:3/4
>     K:F
>     f4 F2 g2 gg gg g4 G2 a2 ba gf
>
>     Breaking it down into smaller segments:
>
>     1. f4 F2
>     2. g2 gg gg
>     3. g4 G2
>     4. a2 ba gf
>
>     Each segment should contain exactly three quarter notes. Let's count the quarter notes in each segment:
>
>     1. f4 F2: 4 (f) + 2 (F) = 6 eighth notes
>     2. g2 gg gg: 2 (g) + 2 (gg) + 2 (gg) = 6 eighth notes
>     3. g4 G2: 4 (g) + 2 (G) = 6 eighth notes
>     4. a2 ba gf: 2 (a) + 2 (ba) + 2 (gf) = 6 eighth notes
>
>     Since the time signature is 3/4, each measure must contain exactly three quarter notes. Therefore, the correct placement of bar lines would be after the first and third segments, as they each contain exactly three quarter notes.
>
>     The correct option is:
>
>     A. L:1/8
>     Q:1/4=120
>     M:3/4
>     K:F
>     | f4 F2 | g2 gg gg | g4 G2 | a2 ba gf |
>
>     So, the answer is \boxed{A}.
>     ```
>
>     These reasoning abilities are foundational skills, essential for LLMs to compose sheet music in a manner akin to human musicians.

---

> > ### Author Response · Authors · 2025-11-20
> > **Author Response (3/3)**
> >
> > 3. The empirical results are unsurprising and offer limited insight. (1) Reading ABC is easier than OMR, expected. (2) A reasoning-tuned model (e.g., DeepSeek-R1) outperforming a large general LLM (e.g., DeepSeek-V3) is also expected. (3) Fine-tuning on music-sheet QA improves in-distribution performance, again expected. More importantly, despite introducing four task categories, the paper provides little analysis about which models work better for which tasks (and why), nor does it disentangle recognition failures from theory/logic errors.
> >
> >
> >     Thank you for your feedback. We agree that our analysis of the results can be significantly strengthened.
> >
> >     Regarding the per-category performance, the data is presented in Figure 6 in our Appendix. The fact that a reasoning-tuned model (e.g., DeepSeek-R1) outperforms a large general LLM (e.g., DeepSeek-V3) is an important finding that validates the reasoning-centric nature of our designed tasks.
> >
> >     Furthermore,  Our framework is designed to separate recognition and reasoning errors, and we agree this needs clearer emphasis.
> >
> >     Each benchmark problem maintains **identical** musical logic, questions, and multiple-choice options across both textual and visual modalities. The only variable is the *representation format*: symbolic ABC notation vs. a rendered image.
> >
> >     This controlled setup enables a clear comparison:
> >
> >     - **Textual QA** measures **pure music reasoning**.
> >     - **Visual QA** assesses **visual recognition and reasoning** together.
> >
> >     This distinction allows us to directly attribute performance differences.
> >
> >     | Model | Textual Modality  | Visual Modality  | Performance Drop |
> >     | --- | --- | --- | --- |
> >     | Gemini-2.5-Pro | 96.19% | 55.44% | **-40.75%** |
> >     | InternVL2.5-78B | 46.06% | 40.44% | **-5.62%** |
> >
> >     This quantitative analysis is a core insight of our paper. It reveals two distinct profiles of model failure:
> >
> >     - For a frontier model like Gemini-2.5-Pro, which demonstrates near-perfect music reasoning in the textual modality (96.19%), the bottleneck is overwhelmingly visual perception. The massive 40-point drop shows that its failure is almost entirely due to challenges in accurately parsing the sheet music image.
> >     - In contrast, for a model like InternVL2.5-78B, the bottleneck is twofold. Its low score on the textual task (46.06%) indicates a fundamental weakness in music reasoning itself. While its visual recognition is more consistent (only a ~6-point drop), its overall poor performance stems from deficiencies in both perception and logic.
> >
> >     This nuanced analysis, enabled by our framework, offers clear directives for future research and model development. We will explicitly frame this cross-modality analysis and the per-category discussion in the revised manuscript to make these contributions clearer.
> >
> >     In addition to the main experiments, our other analyses show that the inherent improvement in sheet music reasoning can enhance both music composition and general mathematical reasoning, which we find particularly interesting.
> >
> > ### Question
> >
> > 1. What is thinking in table 2 and table 3? Do you mean reasoning models? Explain what it means in the paper.
> >
> >
> >     Yes, by "thinking," we refer to a reasoning model. Some reasoning models can operate in a mode where reasoning is either active or inactive. In this paper, the term "thinking" is used to indicate when the reasoning mode is activated.
> >
> >
> > 3. In Section 4.2 the numbers in the main text is different with the numbers in figure 5, which one is correct?
> >
> >
> >     Thank you for pointing this out. After reviewing the results, we found that the correct score is 33.90%, not 33.75%. Detailed Math task performance is in Appendix E. We will revise the main text accordingly.
> >
> >
> > 3. What is the meaning of Section 4.2, showing that the music sheet data can improve reasoning in Math? What is the intuition behind and why is it relevant to the paper?
> >
> >     The intuition behind the cross-domain transfer is based on the shared procedural, rule-based reasoning between music and math tasks.
> >
> >     For example:
> >
> >     - Solving a rhythmic problem involves parsing time signatures and note values, then calculating fractions.
> >     - An interval problem requires calculating the distance between two pitches.
> >     - A chord completion problem uses an algebraic-like formula.
> >
> >     These processes of parsing symbols, recalling rules, and performing calculations mirror the steps needed for many math problems. The model isn’t learning music facts for math; it’s learning the meta-skill of applying rules to symbolic systems.
> >
> > We sincerely appreciate your time and effort in reviewing our submission and our response, and we are happy to address any additional questions or concerns you may have.

---

> ### Comment · Reviewer_u888 · 2025-11-25
>
> Thank you for the timely rebuttal and I appreciate the clarifications and additional results. However, I am still inclined to maintain my current score for two primary reasons:
> 1. **Limited value of the main contribution (programmatic framework + synthetic data).**
> As you note in your response to Reviewer CAkk, “current LLMs are bottlenecked by a lack of training data that is expensive to annotate.” I agree this is an important challenge, but the current framework has modest impact on that bottleneck: it creates QA pairs from text-based ABC notation using nine fixed templates, where most annotations are directly read off or induced from the ABC notation itself. This narrows the significance of the contribution compares to a framework that generates or recovers text-based notation from staff images or from audio/MIDI, thereby addressing the cost of expert annotation and the visual/OMR difficulty that limits practical progress.
>
> 2. **“Music reasoning” and “music theory” are poorly defined and weakly evidenced.**
> The paper leans on these terms, but does not precisely define “musical reasoning,” nor demonstrate tasks that require it. I disagree that mathmatically calculation proves reasoning. The nine question classes are mainly perceptual/identification queries (e.g., note, time signature, interval) whose answers either exist explicitly in ABC or can be computed by straightforward rules. This is quite different from genuinely reasoning (e.g., as in MMMU’s etiology reasoning or other categories). Given that ABC notation encodes the ground-truth symbolic content, much of the challenge reduces to text parsing or rule application rather than substantive musical inference, which is another reason why starting from ground-truth ABC notation inherently limits the scope.

---

> > ### Author Response · Authors · 2025-12-02
> > **Author Response**
> >
> > Thank you for your feedback. We appreciate your concerns; however, we would like to offer a critical clarification regarding the strategic value of our framework and the definition of reasoning in this context.
> >
> > **1. Limited value of the main contribution (programmatic framework + synthetic data).**
> >
> > We respectfully argue that utilizing ABC notation is not a limitation, but a core design feature that unlocks Reinforcement Learning with Verifiable Rewards (RLVR)—a capability previous works have largely overlooked. Our ultimate goal is to build an AI Musician capable of composition, a task that demands rigorous reasoning. By treating music theory as programmatic functions based on text-based scores (ABC), we can establish a modality-invariant reasoning framework. The logical solution to a music problem remains identical whether the input is text, an image, or audio. Consequently, this allows us to synthesize perfectly aligned, verifiable training data across all modalities (Text, Image, Audio/MIDI). This approach effectively bypasses the noise inherent in Optical Music Recognition (OMR) and the high cost of human annotation, providing the precise "ground truth" signals required to train models to reason via RL, rather than merely recognizing symbols.
> >
> > **2. “Music reasoning” and “music theory” are poorly defined and weakly evidenced.**
> >
> > We posit that Musical Reasoning is inherently a domain-specific form of symbolic reasoning. Because sheet music functions as a structured formal language with strict grammatical and mathematical rules, tasks that appear to be simple "calculation" (e.g., interval analysis, rhythmic alignment) are effectively deductive inference steps, analogous to solving mathematical proofs or logic puzzles. We view these tasks as a fundamental case of symbolic reasoning where the model must apply abstract rules to derive correct conclusions. Furthermore, these fundamental operations are non-negotiable prerequisites for high-level musical intelligence. Just as a mathematician cannot solve complex theorems without first mastering arithmetic, an AI cannot perform high-level etiology reasoning or composition without first mastering the "mechanics" of music theory.
> >
> > As a pioneering work, we focused on these low-level rules in music theory to validate the effectiveness of the proposed framework. By demonstrating that models can learn and reason through these fundamental rules, we have laid the necessary groundwork for scaling this verifiable synthesis paradigm to higher-level musical creativity and analysis.

---

### Author Response · Authors · 2025-12-03
**Overview of Our Response and Key Points**

Dear AC,

Thank you for reviewing our rebuttal, and we greatly appreciate the valuable feedback from all reviewers. Their comments have been instrumental in refining our work.

The challenge towards AI musicians lies in the lack of reasoning trace data that accurately simulates a musician's thought process, a task that is prohibitively expensive to annotate manually. While recent advances in RLVR highlight the potential to enhance models' reasoning abilities in this domain, the field is hindered by the absence of verifiable musical questions to apply this technique effectively.

**Our work introduces a new paradigm to address data scarcity in sheet music reasoning.**

By programmatically generating questions based on music theory rules, we transform abstract musical logic into scalable, verifiable computational problems derived from existing sheet music. These problems can then be converted into different modalities by altering the music format. We believe this represents a **groundbreaking** contribution, offering a cost-effective solution to improve LLM and MLLM reasoning in sheet music, ultimately aiding music composition.

While reviewers yS4X, CAkk, and D6f4 acknowledge our novelty and potential, they raised concerns about our focus on low-level music theory. Higher-level music theory and foundational system tools will be addressed in our future work, and we have outlined our research plan and future directions in the rebuttal and the revised paper, which we believe will inspire further exploration in the community.

Additionally, reviewer u888’s concern about the definition and novelty of musical reasoning seems to stem from a misunderstanding in comparing our work to Optical Music Recognition (ORM). The emphasis in our paper is on constructing verifiable problems based on music theory rules from text-based sheet music, which can then be transformed into other modalities.  In this paper, we consider both text and image modalities, and we also plan to extend our research to include MIDI in future studies. Unlike ORM, our primary aim is to enable LLMs to think like musicians.

Finally, we would like to express our sincere gratitude to you and all the reviewers. We hope our work can provide valuable insights and inspiration for the community.

---

### Meta-Review · Area_Chair_8eoH · 2025-12-25

**Summary:**

The recommendation of rejection is mainly based on the fact that many important reviewers' concerns are still not well addressed, detailed below, including "music theory coverage is limited", "do not appear to require genuine reasoning", "composition evaluation is narrow" etc. The two negative reviewers (with ratings of 4 and 2) also confirm that they remain negative after the rebuttal. The AC agrees with reviewers and suggests the authors to take reviewers' suggestions into concern to further improve and polish the paper for resubmission.

**Reviewer Concerns:**

See details below.

**Reviewer Scores:**

***Reviewer u888 is going to maintain the negative rating of 2 as many key concerns are inherently hard to address with rebuttal and the reviewer explicitly mentioned keeping the original score in discussion.***

1. the authors merely scratch the surface in section 2.2 and lots of core details are missing

Partially addressed

2. current questions do not appear to require genuine reasoning

Partially addressed

3. The empirical results are unsurprising and offer limited insight.

Partially addressed

4. Some other minor questions

Mostly addressed

5. "Music reasoning” and “music theory” are poorly defined and weakly evidenced

Partially addressed

***Reviewer CAkk likely to maintain the original rating of 4 as many raised concerns are inherently difficult to address with a rebuttal, and the reviewer explicitly mentioned that "keep my score unchanged".***

1. Music theory coverage is limited.

Partially addressed

2. Performance still far from SOTA baselines

Partially addressed

3. MMMU results close to random guessing.

Mostly addressed

4. Composition evaluation is narrow.

Partially addressed

5. Other minor questions

Mostly addressed

***Reviewer yS4  likely to maintain the original scores of 6 as the reviewer explicitly mentioned "retain my score".***

1. Presentation clarity.

Mostly addressed

2. Dataset only focuses on 4 music theory domains

Partially addressed

3. comparing how the best-performing fine-tuned models perform compared to humans

Mostly addressed

4. Other minor questions

Mostly addressed

***Reviewer D6f4 likely to maintain the original scores of 6 or change the score from 6 to 4 due to some concerns that are not well addressed (2 and 3 below) and influence from other reviewers' concerns***

1. Some assumptions made in this work are not clarified in the paper

Mostly addressed

2. The resulting synthetic data are still fairly simple

Partially addressed

3. limiting the scholarly quality of the paper.

Partially addressed

4. Some other minor questions

Mostly addressed

---

### Decision · Program_Chairs · 2026-01-26

Reject